# ReMix: Towards a Unified View of Consistent Character Generation and Editing

## Abstract

Consistent character generation and editing has made significant strides in recent years, driven by advancements in large-scale text-to-image diffusion models (*e.g.*, FLUX.1) that produce high-fidelity outputs. Yet, few methods effectively unify them within a single framework. Generation-based methods still struggle to enforce fine-grained consistency, especially when tracking multiple instances, whereas editing-based approaches often face challenges in preserving posture flexibility and instruction understanding. To address this gap, we propose **ReMix**, a unified framework for character-consistent generation and editing. It consists of two main components: the ReMix Module and IP-ControlNet. The ReMix Module leverages the multimodal understanding capabilities of MLLM to edit the *semantic content* of the input image, and adapts the instruction features to be compatible with a native DiT backbone. While semantic editing can ensure coherent semantic layout, it cannot guarantee consistency in pixel space and posture controllable. To this end, IP-ControlNet is introduced to coupe with these problems. Specifically, inspired by convergent evolution in biology and by decoherence in quantum systems, where environmental noise induces state convergence, we hypothesize that jointly denoising the reference and target images within a same noise space promotes feature convergence, thereby aligning the hidden feature space. Therefore, architecturally, we extend ControlNet to not only handle sparse signals but also decouple semantic and layout features from reference images as input. For optimization, we establish an $\epsilon$-equivariant latent space, allowing visual conditions to share a common noise space with the target image at each diffusion timestep. We observed that this alignment facilitates consistent object generation while faithfully preserving reference character identities. Through the above design, ReMix supports a wide range of visual-guidance tasks, including personalized generation, image editing, style transfer, and multi-visual-condition generation, among others. Extensive quantitative and qualitative experiments have demonstrated the effectiveness of our proposed unified framework and optimization theory. code: `https://github.com/xxx`.

## 1 Introduction

In recent years, large-scale text-to-image diffusion models (Podell et al., 2023; Esser et al., 2024; Luo et al., 2023; Labs, 2023) have rapidly advanced the generation of high-fidelity images, achieving remarkable visual quality. Simultaneously, the ability to generate controllable and customizable images has gained increasing attention within the research community (Zhang et al., 2023; Peng et al., 2024; Tan et al., 2024; Zhou et al., 2024; Huang et al., 2024). This includes key areas such as face consistency generation (Wang et al., 2024b; Yan et al., 2023; Guo et al., 2024), portrait consistency generation (Ye et al., 2023; Zhou et al., 2024; He et al., 2025; Hu, 2024), posture-controllable portrait generation (Zhang et al., 2023; Zhao et al., 2024; Peng et al., 2024), as well as image editing tasks (Labs et al., 2025; Wu et al., 2025a; Feng et al., 2025; Liu et al., 2025; Xu et al., 2025).

Early approaches, such as ControlNet Zhang et al. (2023) and IPAdapter Ye et al. (2023), introduced coarse-grained control mechanisms by integrating external conditions and guidance signals into the diffusion model architecture. While these pioneering methods provided initial control, they struggled to achieve pixel-level precision while maintaining output fidelity in more complex gener-

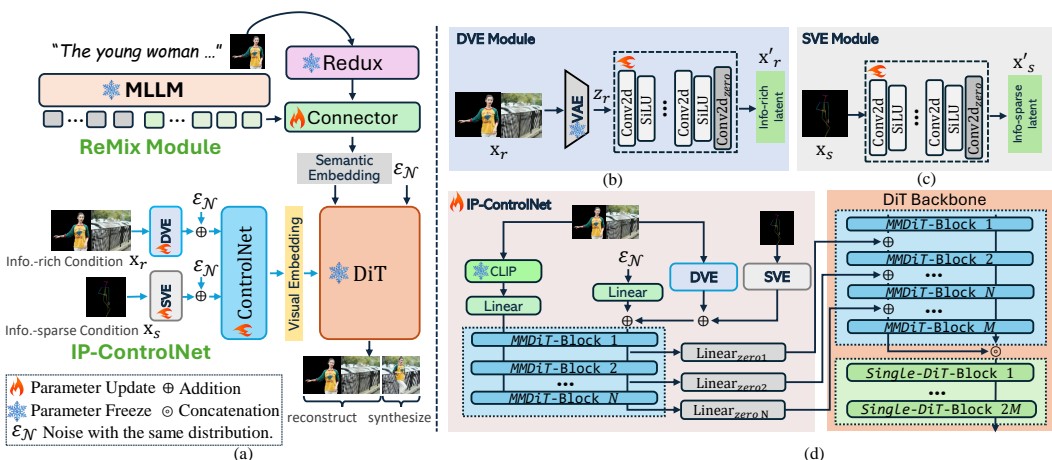

Figure 1: Method Overview. The architecture includes two major components: ReMix Module and IP-ControlNet. **ReMix** Module uses MLLM to edit the semantic content of images, while **IP-ControlNet** controls pixel-level consistent generation by extracting the low-level visual feature.

ation scenarios. Later advancements (Wang et al., 2024b; Han et al., 2024; Zhou et al., 2024; Yan et al., 2023) expanded on these ideas by incorporating more complex control signals, such as face features or human portraits, demonstrating the potential of diffusion models for guided image synthesis. However, despite this progress, significant challenges remain in ensuring consistent character generation. A primary issue is achieving fine-grained consistency, where noise accumulation during the diffusion process often disrupts critical details such as facial expressions, clothing, and other subtle character traits. Essentially, we argue that these models fail to learn consistent latent space representations as noisy progression can disrupt fine-grained dependencies between image components, leading to *semantic misalignments* that hinder the generation of coherent and contextually accurate images across different instances. Recently, image editing methods (Labs et al., 2025; Wu et al., 2025a; Liu et al., 2025) have gained significant momentum in improving spatial consistency. However, a key limitation of these methods is their reliance on strong 2D pixel-level correspondence constraints, which reduces controllability over subject posture and often leads to suboptimal spatial layouts. Consequently, effectively integrating character-consistent generation and editing remains an open challenge.

To tackle the above problem, as shown in Figure 1a, we propose a unified framework for character-consistent generation and editing, consisting of two main components: the ReMix Module and IP-ControlNet. The ReMix Module integrates a pre-trained MLLM Bai et al. (2023) that accepts both text instructions and reference images. A connector is employed to refine the semantic features of Redux Labs et al. (2025) using the MLLM's instruction features, thereby adapting the MLLM's output hidden states to the native DiT backbone without requiring DiT fine-tuning. This design substantially reduces training costs while preserving DiT's native image generation capabilities, in contrast to Liu et al. (2025), which retrain DiT to adapt to MLLM features. While semantic editing is effective in producing semantically consistent results, it cannot guarantee fine-consistency and controllability in pixel level. To overcome this limitation, we introduce IP-ControlNet, an enhanced ControlNet AI (2024) architecture designed to enforce pixel-space consistency during generation and editing. Specifically, for any information-rich visual cues (*e.g.*, human portrait), we decompose the visual conditions into semantic guidance flows captured by CLIP Kim et al. (2022) and visual guidance maps extracted by the proposed Dense Visual Encoder (DVE) module. In contrast, for any information-sparse visual cues (*e.g.*, human pose), we apply nonlinear mapping directly through the proposed Sparse Visual Encoder (SVE) module. Finally, the extracted visual features undergo joint attention via the MMDiT module and are fused into the DiT backbone through a recurrent feature-fusion scheme, following AI (2024). However, we observed that independently injecting conditional signals did not achieve the desired pixel-level consistency, particularly when multiple visual controls, such as portrait, background, and pose were present. We hypothesize that this limitation stems from the *semantic misalignment* problem mentioned earlier. Inspired by convergent

evolution in biology Losos (2011) and decoherence in quantum systems Zurek (2003), where environmental noise drives state convergence, we propose an alignment method by enforcing feature convergence through shared-space denoising. This promotes feature convergence and aligns the hidden feature space. Unlike existing methods (Tan et al., 2024; Zhang et al., 2025; Wu et al., 2025c), which directly inject pixel features into the target noise space for joint-attention, our solution is models the dependencies between conditional inputs and target outputs in an $\epsilon$-equivariant feature space, aims to simulation an "homogenization" effect.

The key contributions of this paper are as follows:

- We propose ReMix, a unified framework for character-consistent generation and editing that integrates semantic adaptation via the ReMix Module and pixel-level control via IP-ControlNet. Offering a novel perspective for achieving high-fidelity image editing.

- We introduce an $\epsilon$-equivariant alignment strategy that denoises reference and target images within a shared noise space, promoting feature convergence and achieving fine-grained character consistency.

- Our method is efficient, it achieves image Generation and Editing without retraining the DiT backbone, reducing training cost while preserving its native generation capability.

## 2 RELATED WORK

### 2.1 CONDITIONAL CONTROL

**Semantic-Level Control** Semantic control in diffusion models has seen significant progress. BoundaryDiffusion Zhu et al. (2024) offers a lightweight, unified single-step operation for semantic control without additional training costs, identifying semantic boundaries without learning. DiffusionCLIP Kim et al. (2022) and Astrp Kwon et al. (2022) also enable semantic control but require significant training time. Recent methods like SDG Liu et al. (2023) and TtfDiffusion Yu et al. (2025) offer learning-free fine-grained control: SDG injects semantic signals for text- and image-based control, while TtfDiffusion discovers semantic directions within pre-trained models during denoising. In addition, methods like Prompt-Free Diffusion Xu et al. (2024) and ViCo Hao et al. (2023) focus on feature modulation but often require subject-specific optimization. While semantic-level control is useful for many tasks, achieving precise semantic understanding and semantic modification remains a challenge. **Pixel-Level Control** Achieving pixel-level control in diffusion models requires spatial alignment while preserving generative diversity. Early methods like ControlNet Zhang et al. (2023) inject spatial cues but often overfit with complex inputs. ControlNet++ Li et al. (2024a) improves alignment using cycle consistency, and ControlNet-XS Zavadski et al. (2024) enhances control fidelity through more frequent interactions. However, balancing precision and diversity remains challenging. UniControl Zhao et al. (2024) and T2I-Adapter Mou et al. (2024) unify multi-modal guidance but struggle with pixel-wise constraints in stochastic sampling. In fact, pixel-level control usually means being subject to stronger spatial consistency constraints and often showing more artifacts.

### 2.2 CHARACTER CONSISTENCY IMAGE SYNTHESIS

Early methods like DreamBooth Ruiz et al. (2023) and Textual Inversion Gal et al. (2022) align outputs with reference subjects but struggle with generalization. Encoder-based approaches such as IPAdapter Ye et al. (2023) and PhotoMaker Li et al. (2024b) improve identity consistency through cross-attention, but face challenges with complex poses. Training-free methods like Custom Diffusion Kumari et al. (2023) and E4T Gal et al. (2023) adapt pre-trained models but trade off identity preservation for flexibility. Hybrid frameworks Kim et al. (2023) combine sketch guidance with reference-based diffusion, yielding high-quality results but limited cross-domain generalization. Recent work (Tan et al., 2024; Zhang et al., 2025; Wu et al., 2025c) with the open-source FLUX.1 Labs (2023) has made progress in spatial consistency, but they inherently rely on strong pixel-level correspondences, which restrict pose flexibility and often lead to unnatural or rigid spatial layouts. Moreover, when multiple conditions (*e.g.*, pose, background, and identity) are combined, the lack of feature-level alignment can result in semantic misalignment and degraded consistency. In short, while generation methods can produce visually convincing characters, they struggle to maintain

162
163
164

consistent identity during editing, and editing methods cannot robustly generalize across diverse generation settings.

165
166

### 2.3 IMAGE EDITING WITH DIFFUSION MODELS

167
168
169
170
171
172
173
174
175
176
177
178
179
180
181
182
183
184

Diffusion-based image editing has emerged as a powerful paradigm for manipulating visual content under semantic guidance. Early methods such as SDEdit Meng et al. (2021) and Prompt-to-Prompt Hertz et al. (2022) leveraged the generative trajectory of diffusion models to modify images by partially resampling latent states while preserving structural coherence. These approaches demonstrated strong editability but often suffered from limited controllability, particularly when handling complex spatial layouts or fine-grained semantics. Subsequent works sought to enhance edit precision through explicit conditioning. InstructPix2Pix Brooks et al. (2023) introduced instruction-tuned models capable of performing text-driven edits by aligning with human-written editing instructions. Kontext Labs et al. (2025) incorporated multimodal instruction alignment into the editing pipeline, improving edit controllability through joint reasoning over text and visual references. Similarly, STEP-1x-Edit (Liu et al., 2025) and Qwen-Image Wu et al. (2025a) retrains a DiT backbone to adapt multimodal features from large language models (MLLMs), enabling higher-fidelity edits with stronger instruction alignment. Despite these advances, such retraining introduces high computational costs and compromises the model's native generative capabilities. Moreover, existing methods focus primarily on editing, with limited integration into character-consistent generation frameworks. These gaps motivate our work: a unified, feature-aligned framework that achieves character-consistent generation and editing within a single model, without sacrificing efficiency or flexibility.

185
186
187
188
189
190
191
192
193
194
195
196
197

### 3 METHOD

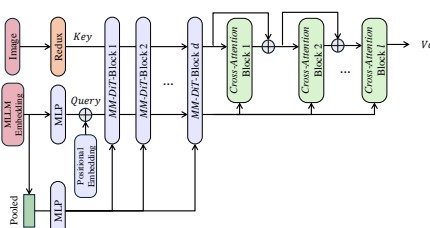

(a) Detailed structure of the connector.  (b) ReMix Module training process.

198
199
200

Figure 2: Overview of Semantic Editing Pipeline. The ReMix module implements semantic editing of Redux Black Forest Labs (2024) features through a learnable Connector.

201
202
203

### 3.1 SEMANTIC EDITING

204
205
206
207
208
209
210
211
212
213

In image generation, recent studies (Liu et al., 2025; Wu et al., 2025a) have shown that integrating embeddings from Multimodal Large Language Models (MLLMs) can substantially improve the semantic understanding of Diffusion Transformers (DiTs). However, these approaches retrain the entire DiT end-to-end to adapt MLLMs, which is computationally expensive and often degrades its native generative capabilities. Therefore, a key challenge lies in how to effectively adapt DiT to MLLM embeddings while keeping its parameters frozen? Building on advances in image variation modeling, particularly Redux (Black Forest Labs, 2024), we observe that introducing a lightweight instruction editor provides an elegant solution to this problem. To be specific, we adopt FLUX.1-dev Black Forest Labs (2024) as the DiT backbone, where the proposed ReMix module enrich the original T5-XXL Raffel et al. (2020) instruction embeddings with refined multimodal embeddings derived from the MLLMs.

214
215

**MLLM** We employ Qwen2.5-VL-7B-Instruct Qwen Team (2025) as the base MLLM. Given the text prompt and reference image, we use the last hidden layer state of the model's output as the editing instruction feature, which is then fed into the Connector module.

**Connector** The Connector aligns MLLM outputs with Redux features extracted from the reference image, as illustrated in Figure 2a. Architecturally, it adopts a dual-stream design: (1) the Redux features derived from the input image form the *key* stream, while (2) the multimodal MLLM embeddings, after nonlinear mapping and positional encoding, constitute the *query* stream. The *key* and *query* features are first processed through multiple MMDiT blocks, where pooled MLLM embeddings provide the global vector input to enhance semantic alignment. The resulting representations are then passed through several cross-attention layers, which modulate the *key* stream to produce a refined *value* stream tailored for integration into the downstream DiT backbone. The process can be summarized as:

$$\text{value} = \psi(\text{query}, \text{key}) \tag{1}$$

where $\psi$ represents the Connector module.

**Redux** As shown in Figure 2b, Redux employs the SigLIP encoder to extract semantic representations from images, followed by a learnable MLP layer that adapts these features to the FLUX.1 backbone. The adapted Redux features are subsequently fused with T5 embeddings, providing complementary semantic and textual guidance to steer the diffusion process. Simply put, Redux extracts the semantic features of the image and then uses it as the text stream for FLUX.1.

**ReMix Training** Since the training process is decoupled, only the Connector requires optimization at this stage. As illustrated in Figure 2b, we adopt a mean squared error (MSE) loss to align the Connector's output *value* with the Redux features *target* extracted from the ground-truth image:

$$\mathcal{L}_{\text{MSE}} = ||\text{value} - \text{target}||_2^2. \tag{2}$$

Notably, the DiT backbone does not need to be involved throughout this stage, making the training procedure highly efficient.

### 3.2 Consistent Character generation

To achieve fine-grained character consistency and flexible layout control, we introduce IP-ControlNet (shown in Figure 1d), a plug-in conditioning module that decouples semantic and spatial information from multi-granularity visual inputs. Given one or more reference images and an optional sparse conditional image, IP-ControlNet processes these inputs through two specialized encoders: a Dense Visual Encoder (DVE) and a Sparse Visual Encoder (SVE).

**DVE** As illustrated in Figure 1b, the DVE module handles highly informative visual prompts, such as full-character or facial reference images. These inputs are rich in both semantic and spatial detail and require careful preservation during conditioning. Given a dense input image $\mathbf{x}_r \in \mathbb{R}^{H \times W \times 3}$, we first obtain a compressed latent feature $\mathbf{z}_r \in \mathbb{R}^{h \times w \times c}$ via the VAE encoder in FLUX.1. To refine and adapt these features for diffusion conditioning, we apply a lightweight, cascaded nonlinear transformation $\psi_{\text{DVE}}$:

$$\mathbf{x}'_r = \psi_{\text{DVE}}(\mathbf{z}_r), \tag{3}$$

where $\psi_{\text{DVE}}$ consists of 6 Conv2D layers (without downsampling, activated by SiLU), totaling only 0.3M parameters to ensure minimal computational overhead. The transformed latent $\mathbf{x}'_r$ is injected into the DiT noise prediction stream via a zero-initialized convolutional adapter $\mathcal{C}_0$:

$$\epsilon'_t = \epsilon_t + \alpha \cdot \mathcal{C}_0(\mathbf{x}'_r), \tag{4}$$

where $\epsilon_t$ is the base DiT prediction at timestep $t$, and $\alpha$ controls the injection strength. To further enhance semantic guidance, we extract global visual embeddings from CLIP Kim et al. (2022) using $\mathbf{x}_r$, and concatenate them with text embeddings from T5-XXL. These combined features form the text stream input to MMDiT, as we observed that this approach can improve the quality of generated content.

**SVE** As illustrated in Figure 1c, SVE is an optional module tailored for low-information spatial cues, such as pose skeletons or edge maps, which provide minimal semantic context but define strong layout priors. Unlike dense inputs, sparse inputs do not go through a VAE, avoiding unnecessary compression. Given a sparse conditional image $\mathbf{x}_s \in \mathbb{R}^{H \times W \times 3}$, we extract layout features directly via a convolutional projector $\psi_{\text{SVE}}$:

$$\mathbf{x}'_s = \psi_{\text{SVE}}(\mathbf{x}_s), \tag{5}$$

where $\psi_{\text{SVE}}$ shares a similar structure to $\psi_{\text{DVE}}$ but includes strided convolutions for downsampling. The layout features are injected into the noise stream via a separate adapter $\mathcal{C}_1$. Then, the formula 4 can be rewritten as:

$$\epsilon'_t = \epsilon_t + \alpha \cdot \mathcal{C}_0(\mathbf{x}'_r) + \beta \cdot \mathcal{C}_1(\mathbf{x}'_s), \tag{6}$$

where $\beta$ modulates the contribution of sparse control signals. This dual-branch design allows sparse layout signals to guide global structure, while dense semantic cues ensure identity preservation and fine detail.

Although the decoupling of semantic and layout features via IP-ControlNet improves conditional representation, we observe that visual consistent deteriorates during the denoising, especially when multiple visual conditions (*e.g.*, portrait and background) are fused. This signal dilution leads to inconsistent pixel-level guidance, weakening both character fidelity and spatial alignment. To address this, we introduce a concept of $\epsilon$-equivariant latent space, *i.e.*, it produces effects similar to convergent evolution in biology.

**$\epsilon$-equivariant Optimization** $\epsilon$-equivariant optimization explicitly enforces latent space alignment and condition-aware generation by jointly optimizing two objectives: conditional reconstruction and target image synthesis. This design encourages the model to maintain coherent spatial signals throughout the denoising trajectory. **Diffusion Loss** Given a set of $N$ info-rich visual prompts $\{\mathbf{x}_r^i \in \mathbb{R}^{H \times W \times 3}\}_{i=1}^N$, we horizontally concatenate them into a single composite tensor: $\mathbf{X}_r \in \mathbb{R}^{H \times (N \cdot W) \times 3}$, which is then encoded by the DVE module into a structured latent feature map $\mathbf{Z}_r \in \mathbb{R}^{h \times (N \cdot w) \times c}$. This latent preserves both intra-image and inter-condition spatial dependencies. After that, our optimization goal becomes: (1) recover the original visual prompt $\{\mathbf{x}_r^i\}$ from the shared latent $\mathbf{Z}_r$:

$$\mathcal{L}_{recon} = \sum_{i=1}^N \mathbb{E}_{t,\epsilon}\big[||\mathbf{x}_r^i - D_\theta(\mathbf{Z}_r^{(t,\epsilon)}, t)||_2^2\big], \tag{7}$$

where $D_\theta$ denotes the denoiser and $\mathbf{Z}_r^{(t,\epsilon)}$ is the noised latent at timestep $t$, and (2) synthesize the final output image $\mathbf{y}$ conditioned on $\mathbf{Z}r$:

$$\mathcal{L}_{gen} = \mathbb{E}_{t,\epsilon}\big[||\epsilon - \epsilon_\theta(\mathbf{y}_t, t|\mathbf{Z}_r)||_2^2\big], \tag{8}$$

By optimizing these two objectives jointly, the model is encouraged to treat the reference features and generation targets as denoising-equivalent, promoting feature convergence in a shared noise space, called $\epsilon$-equivariant latent space in this paper. This latent alignment is essential for ensuring cross-instance consistency, even when visual prompts vary in density or semantic richness. To simplify the implementation, we unify the dual objectives of reconstruction and generation into a simplified flow matching loss formulation. Let $\epsilon \in \mathbb{R}^{h \times (N \cdot w) \times c}$ denote the ground-truth noise, and $\mathbf{Z}_r \in \mathbb{R}^{h \times (N \cdot w) \times c}$ represent the concatenated latent of visual prompts. The DiT backbone predicts the noise residual $\epsilon_\theta$, which is optimized via:

$$\mathcal{L}_{equ} = \mathbb{E}_{t,\epsilon}\big[||\epsilon_\theta(\mathbf{y}_t, t|\mathbf{Z}_r) - (\epsilon - \mathbf{Z}_r)||_2^2\big]. \tag{9}$$

where $\mathbf{Z}_r$ acts as a reconstruction prior, steering the model to recover input conditions while denoising $\mathbf{y}_t$. **ID Loss** For human subjects, we incorporate PuLID Guo et al. (2024) into both IP-ControlNet and DiT. However, we empirically observe the model gradually "forgets" identity cues as it prioritizes denoising fidelity. To mitigate this, we introduce an identity-consistency loss $\mathcal{L}_{id}$ that explicitly constraints facial attributes across generations. Let $\mathbf{z}_{gen}$ and $\mathbf{z}_{ref}$ be the face embeddings extracted from the generated image and reference image using ArcFace Deng et al. (2019), respectively. Then, the ID loss can be defined as follows:

$$\mathcal{L}_{id} = 1 - \frac{\mathbf{z}_{gen}\mathbf{z}_{ref}}{||\mathbf{z}_{gen}||_2||\mathbf{z}_{ref}||_2} \tag{10}$$

which maximizes cosine similarity between identity features. **Total Loss** The final total loss is defined as:

$$\mathcal{L}_{total} = \mathcal{L}_{equ} + \lambda \mathcal{L}_{id}, \tag{11}$$

where $\lambda = 0.2$ balances the objectives. **Inference** During inference, to reconstruct a consistent conditional image, we balance the preservation of the original image by skipping part of the diffusion process. Specifically, we construct the initial noise for condition image according to the intensity coefficient $t \in [0, 1]$: $X_r^{\mathcal{N}} = t \cdot \epsilon + (1 - t) \cdot X_r$, where $\epsilon \sim \mathcal{N}(0, I)$, the smaller the $t$, the closer to the original image. Here we set $t = 0.5$. **Positional Encoding** To prevent positional conflicts, we assign the generated region indices sequentially after those of the reconstructed region, with the upper-left corner of the generated image following the lower-right corner of the reconstructed one, ensuring smooth spatial indexing.

Table 1: Quantitative comparison for human/subject-centric image generation. The best results are in **bold** and the second best results are underlined. * indicates post-training using corresponding training dataset used in this paper.

| Method | Human-Centric | | | | Subject-Centric | | |
|---|---|---|---|---|---|---|---|
| | CLIP-I↑ | DINO↑ | CLIP-T↑ | ID-Sim.↑ | CLIP-I↑ | DINO↑ | CLIP-T↑ |
| Textual Inversion Gal et al. (2022) | - | - | - | - | 78.0 | 56.9 | 25.5 |
| BLIP-Diffusion Li et al. (2023) | 71.4 | 50.8 | 24.3 | - | 75.5 | 55.6 | 27.4 |
| SSR-Encoder Zhang et al. (2024) | 77.4 | 61.0 | 31.0 | - | - | - | - |
| DreamBooth Ruiz et al. (2023) | - | - | - | - | 81.2 | 69.6 | 30.6 |
| IP-Adapter (FLUX.1)* Ye et al. (2023) | 80.6 | 60.5 | 31.2 | 0.1 | 84.3 | 64.4 | 35.9 |
| OmniControl* Tan et al. (2024) | 77.4 | 62.8 | 31.0 | 0.2 | 85.7 | **70.3** | 35.8 |
| PuLID (FLUX.1) Guo et al. (2024) | 78.1 | 57.3 | 31.3 | 0.7 | - | - | - |
| **ReMix (ours)*** | **87.3** | **71.0** | **32.3** | 0.7 | **86.7** | 70.2 | **36.3** |

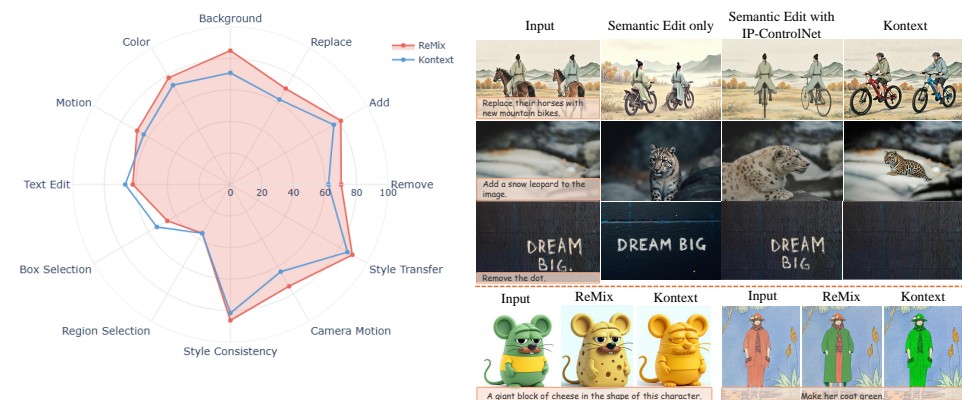

(a) Qualitative comparison.    (b) Quantitative comparison.

Figure 3: (a) Qualitative comparison results on Kontext-Bench1K Labs et al. (2025). (b) Quantitative visualization results (See supplementary material for more).

## 4 EXPERIMENT

### 4.1 EXPERIMENTAL SETUP

**Dataset Setting**   We evaluate ReMix across two primary tasks: human/subject-centric generation and image editing. For human-centric generation, we train on an internal dataset containing 180,764 image pairs from 13,498 unique human identities. Following EMMA Han et al. (2024), we construct a one-to-many test set comprising 32 portraits, each paired with four prompts, and generate five images per prompt to evaluate identity consistency. For subject-centric generation, we train on the Subjects200K dataset Tan et al. (2024) and evaluate on the DreamBooth benchmark Ruiz et al. (2023), producing four stochastic samples per prompt. For image editing, except to the paired data mentioned above, we also mixed the OmniEdit Wang et al. (2024a) and OmniGen2 Wu et al. (2025b) datasets, trained the semantic editing module from scratch, and then evaluated it on the Kontext-Bench1K Labs et al. (2025), a newly released comprehensive benchmark for image editing.

**Model Setting**   For efficiency, we configure IP-ControlNet with $N = 4$ blocks, and set the Connector dimensions to $d = 4$ and $l = 8$. Training is performed with a learning rate of $1 \times 10^{-5}$ and a batch size of 16. The ReMix Module is trained for 1.5M iterations across all datasets. For IP-ControlNet, we freeze the ReMix Module parameters and first conduct a warm-up phase of 80K iterations using a one-to-one[1] data setting, followed by 600K iterations with a one-to-many set-

---

[1]Forming pairs with the image itself.

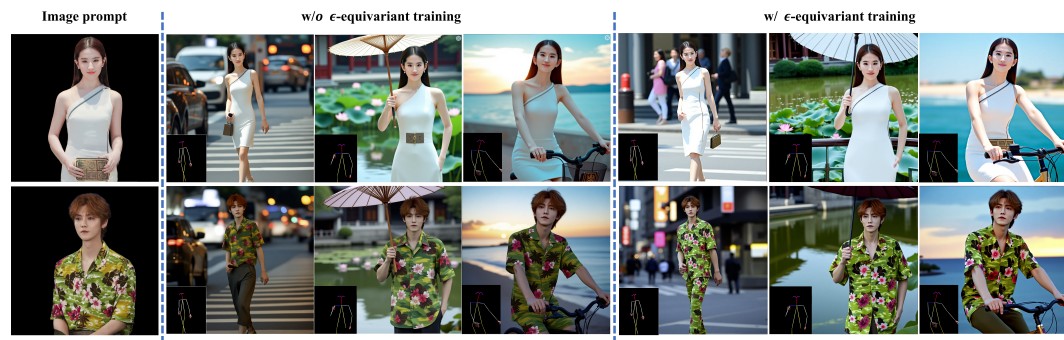

Figure 4: Effect of $\epsilon$-equivariant optimization. Middle: 500K iter. standard one-to-many setting; Right: last 100K iter. $\epsilon$-equivariant training.

Table 2: Ablation of $\epsilon$-equivariant optimization.

| Variant | CLIP-I↑ | DINO↑ | CLIP-T↑ |
|---|---|---|---|
| vanilla | 86.4 | 68.9 | 30.7 |
| w/ $\epsilon$-equivariant | **87.3** | **71.0** | **32.3** |

Table 3: MLLM embedding *vs*. T5 embedding.

| Variant | CLIP-I↑ | DINO↑ | CLIP-T↑ |
|---|---|---|---|
| T5 embedding | 84.1 | 70.7 | 31.0 |
| MLLM embedding | **87.3** | **71.0** | **32.3** |

Table 4: Hyperparameters in Connector. The gray rows represent the selected hyperparameters in this paper.

| MMDiT Blocks $d$ in Connector ($l = 8$) | | | |
|---|---|---|---|
| Configuration | #params | CLIP-I | CLIP-T |
| $d = 2$ | $\sim$700M | 85.7 | 31.7 |
| $d = 4$ | $\sim$1.5B | 87.3 | 32.3 |
| $d = 6$ | $\sim$2.0B | **88.6** | **33.3** |
| Cross-Attention Blocks $l$ in Connector ($d = 4$) | | | |
| $l = 4$ | $\sim$100+M | 86.1 | 32.1 |
| $l = 8$ | $\sim$200+M | 87.3 | 32.3 |
| $l = 12$ | $\sim$400+M | **87.6** | **33.7** |

ting. All experiments are trained on $8 \times$ H800 GPUs. To improve training efficiency, the proposed $\epsilon$-equivariant optimization is applied in the last 100K iterations under the one-to-many data setting.

### 4.2 MAIN RESULTS

**Consistent Character Generation** Table 1 presents the quantitative comparison between our method and state-of-the-art approaches (See Appendix for qualitative comparison). *Human-Centric Generation* ReMix outperforms existing methods in visual alignment, achieving improvements of +6.7% (CLIP-I) and +8.2% (DINO). This improvement in visual consistency is primarily attributed to IP-ControlNet and $\epsilon$-equivariant optimization, as demonstrated by our ablation experiments. *Subject-Centric Generation* ReMix also consistently achieves superior results in this subtask, with +1.1% improvement over OmniControl and +0.4% improvement over DreamBooth. Notably, ReMix strikes an optimal balance between identity retention (CLIP-I/DINO) and textual alignment (CLIP-T, +1.4%), showcasing its robust performance in both preservation and alignment tasks.

**Image Editing** We benchmark our model against the open-source state-of-the-art Kontext Labs et al. (2025). To enable a comprehensive evaluation, we categorize the Kontext-Bench1K dataset into 12 editing types (*e.g.*, remove, add) and conduct comparisons across all categories. As shown in Figure 3a, our model delivers consistently strong results, with notable gains on most common tasks such as Add, Remove, Replace, Background and so on. However, ReMix did not show significant improvements in text editing and some less common tasks (*e.g.*, bounding box selection and region-based editing), which we attribute to a lack of relevant data. Since ReMix performs edits at the semantic level, it achieves a deeper scene understanding compared to the T5-based features in Kontext, producing more coherent layouts and natural visual compositions, as shown in Figure 3b.

### 4.3 ABLATION STUDY

**$\epsilon$-equivariant Optimization** We conducted a controlled ablation study to assess the impact of this training strategy. As shown in Table 2, removing $\epsilon$-equivariant optimization results in a per-

formance degradation across all metrics, with textual alignment (CLIP-T) decreasing by -4.9% and spatial coherence (DINO) decreasing by -2.8%. Additionally, Figure 4 visually demonstrates that $\epsilon$-equivariant optimization can enhance fine-grained character consistency, especially spatial layout (maintain the skirt silhouette) and semantic correction (fix color attributes). These results confirm the importance of learning in a shared feature space, specifically the $\epsilon$-equivariant latent space for diffusion models, to ensure both semantic accuracy and spatial consistency.

**ReMix Module**   As illustrate in Table 3, an interesting observation emerges in the character-consistent generation task: replacing T5 with MLLM embeddings substantially improves the CLIP score, while the DINO score remains largely unaffected. This suggests that MLLM embeddings enrich the semantic space of instruction representation, improving text–image alignment as captured by CLIP. In contrast, DINO focuses purely on visual sim-

Table 5: Visual Instruction Decoupling

| Variant | CLIP-I↑ | DINO↑ | CLIP-T↑ |
|---------|---------|-------|---------|
| vanilla | **87.3** | **71.0** | **32.3** |
| w/o DVE | 77.6 | 55.1 | 30.9 |
| w/o CLIP | 85.6 | 65.0 | 30.6 |

ilarity; since IP-ControlNet already enforces pixel-level consistency, the added semantic cues from MLLM embeddings have minimal effect on DINO. Moreover, Table 4 analyzes the impact of Connector capacity. Balancing efficiency and performance, we set $d = 4$ and $l = 8$ in our final design.

**IP-ControlNet**   *Visual Instruction Decoupling* The performance degradation in ablated variants (Table 5) highlights the distinct roles of DVE module and CLIP guidance in our framework. As can be seen, removing the DVE module leads to an $-11.1\%$ drop in CLIP-I and $-22.3\%$ in DINO score, demonstrating that VAE latent is critical for preserving fine-grained identity features and geometric consistency. Its absence forces the model to rely on global semantic embeddings, which lack pixel-level alignment. While the absence of CLIP guidance in IP-ControlNet seems to have little impact on both CLIP-I and CLIP-T, we speculate that this is because the ReMix module itself provides sufficient semantic guidance. *Parameters* Table 6 reports the effect of varying MMDiT blocks and Conv2D layers in IP-ControlNet. To ensure the best trade-off between accuracy and efficiency, we adopt $N = 4$ and 6 Conv2D layers as the default setting.

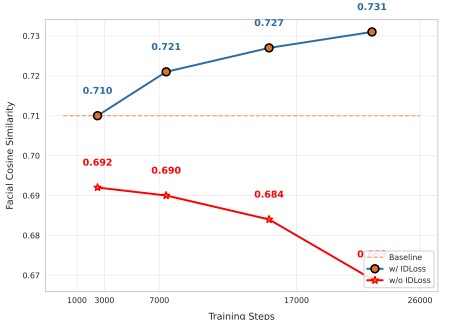

Figure 5: ID Similarity vs. Training Steps.

Table 6: Hyperparameters in IP-ControlNet.

| **MMDiT Blocks $N$ in IP-ControlNet** | | | |
|---------------------------------------|---------|--------|--------|
| Configuration | #params | CLIP-I | DINO |
| 2 MMDiT Blocks | 743M | 83.82 | 65.61 |
| 4 MMDiT Blocks | 1.4B | 87.32 | 70.99 |
| 6 MMDiT Blocks | 2.1B | **88.05** | **71.12** |
| **Number of Conv2D Layers in SVE and DVE** | | | |
| 4 Conv2D Layers | 0.2M | 86.88 | 70.13 |
| 6 Conv2D Layers | 0.3M | **87.32** | 70.99 |
| 8 Conv2D Layers | 0.4M | 87.28 | **71.01** |

**ID Loss**   This experiment reveals critical phase transitions in identity preservation during diffusion training. As shown in Figure 5, without the $\mathcal{L}_{id}$ constraint, face similarity (measured by ArcFace similarity Deng et al. (2019)) decays exponentially after 17k training steps. This because vanilla diffusion processes suffer from identity dissipation in low-noise regimes.

## 5   SUMMARY

In this work, we introduced ReMix, a unified framework for character-consistent image generation and semantic image editing. Unlike existing approaches, ReMix leverages MLLM embeddings through a learnable Connector module, preserving the generative power of the frozen DiT while enabling flexible instruction adaptation. We demonstrated that MLLM embeddings significantly enhance text–image alignment without sacrificing visual fidelity. ReMix achieves competitive or superior performance across human/object-centric generation and diverse editing tasks, particularly excelling in manipulations requiring deep semantic understanding . With its modular design, efficient training, and semantic-level understanding, ReMix provides a practical and generalizable solution for bridging generation and editing within a single consistent framework.

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
