# SUPPLEMENTARY FOR "ReMix: Towards a Unified View of Consistent Character Generation and Editing"

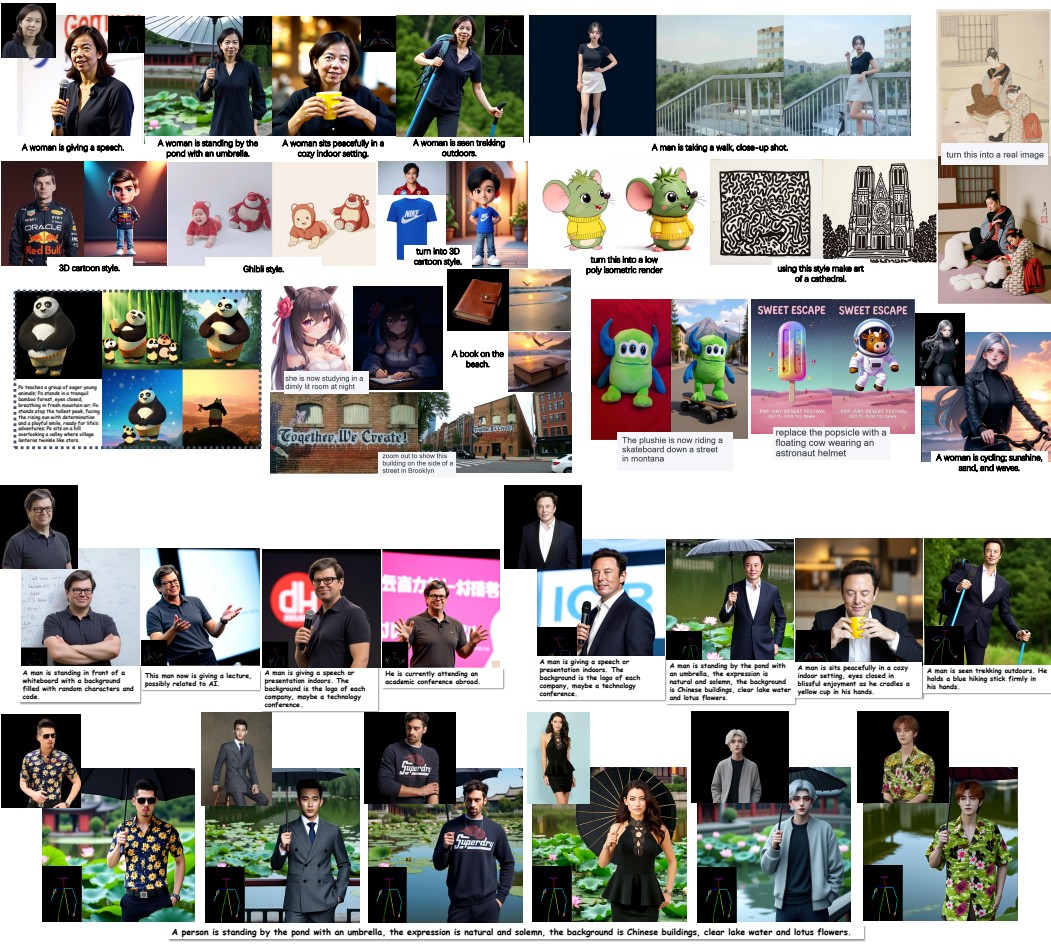

Figure 1: Examples generated by ReMix. The framework supports a wide range of multimodal synthesis tasks, including personalized generation, image editing, layout-consistent synthesis, style transfer, multi-condition generation, and narrative-driven story visualization, *etc*.

## A OVERVIEW

The supplementary materials are organized as follows. We first present additional qualitative results on image editing in Section B. Next, we provide further examples of character-consistent generation, including both human- and subject-centric cases in Section C.1, as well as other image-generation-based tasks such as multi-condition generation in Section C.2, character-consistent style transfer in Section C.3, story visualization in Section C.4, and compositional image generation in Section C.5.

Finally, we include a statement clarifying the limited role of LLMs in assisting with the stylistic refinement of this work in Section D.

# B IMAGE EDITING

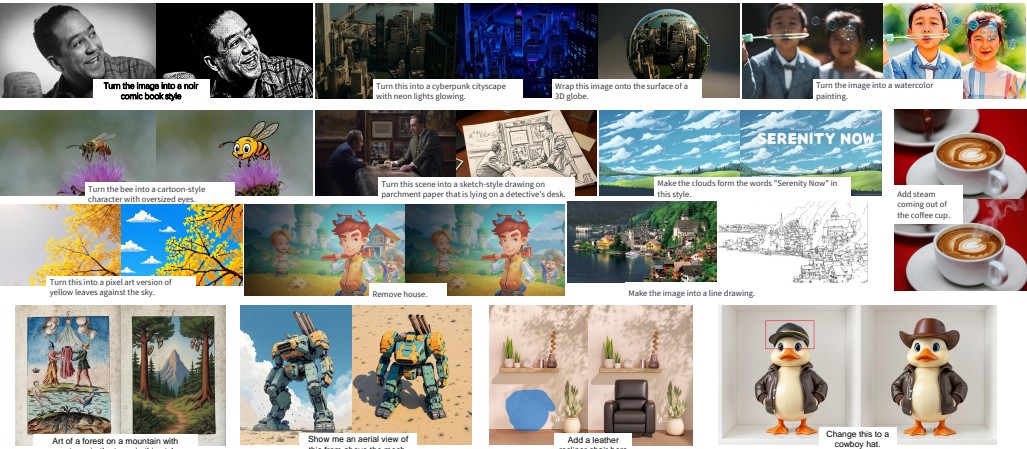

(a) Results of consistent image editing.

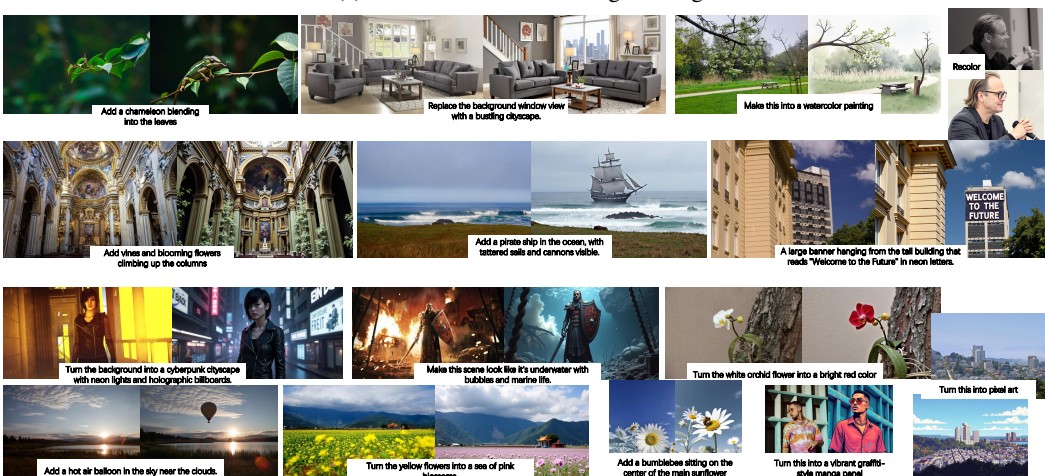

(b) Results of semantic image editing.

Figure 2: Qualitative Results of ReMix on Image Editing.

**Qualitative Results of ReMix on Image Editing** Figure 2a presents additional editing results, demonstrating ReMix's support for a wide range of editing types. Beyond basic operations such as adding or removing elements, ReMix also enables more advanced tasks like region-based and box-based editing. The qualitative results highlight that our framework is not only competitive with existing approaches but also highly efficient: ReMix achieves strong editing performance with minimal training cost, requiring only two lightweight modules trained separately—a Connector for semantic editing and an IP-ControlNet for pixel-level consistency.

**Training-free Semantic Editing with the ReMix Module** Owing to its flexible, modular design, ReMix Module can be seamlessly integrated with nearly all models in the FLUX.1 family. As shown in Figure 2b, we directly concatenate the ReMix output with the T5 embedding of the text-to-image model FLUX.1-Dev, effectively extending it into a text-guided image editing framework. The generated results demonstrate that ReMix accurately interprets both editing instructions and image content, producing high-fidelity edits without requiring any retraining of the DiT backbone.

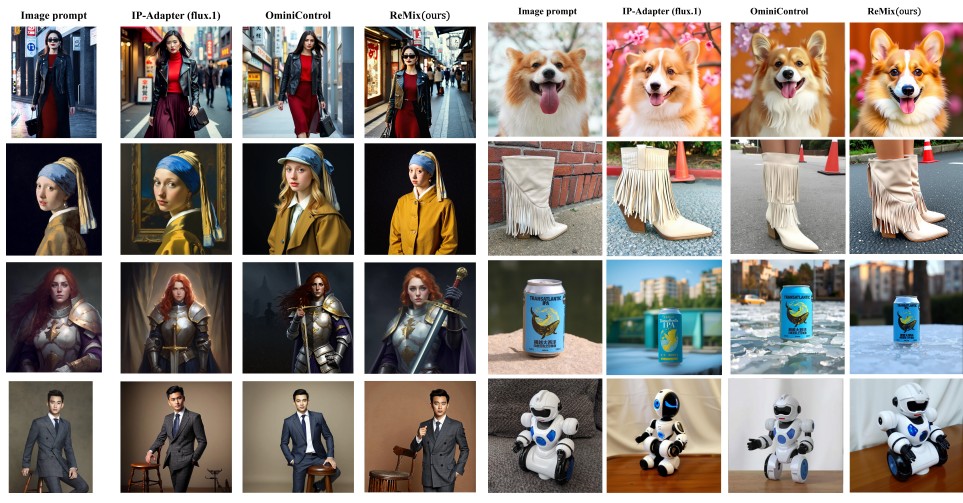

(a) Human-Centric.          (b) Subject-Centric.

Figure 3: Comparison with different DiT models.

Beyond FLUX.1, ReMix generalizes to models that rely on T5 Raffel et al. (2020) or BERT Devlin et al. (2019) for instruction embeddings extracting, requiring only lightweight task-specific fine-tuning. Importantly, *it provides a resource-efficient solution for injecting MLLM embeddings into DiT models*, significantly enhancing their semantic understanding while preserving efficiency.

## C   CONSISTENT CHARACTER GENERATION

In this experiment, we used the same text description and seed to verify the stability of character-consistent image generation models, such as IP-Adapter Ye et al. (2023) and OmniControl Tan et al. (2024). We evaluate the reconstruction ability of the model by comparing fine-consistency in pixel space.

### C.1   QUALITATIVE COMPARISON

Our method excels in preserving fine-grained identity features, particularly in garment texture fidelity, as illustrated in Figure 3a and Figure 3b. These results underscore the limitations of purely CLIP-based feature alignment methods (*e.g.*, IP-Adapter Ye et al. (2023)) when detailed restoration is required. By incorporating low-level visual priors through the DVE module and leveraging the $\epsilon$-equivariant optimization, ReMix enhances spatial consistency and establishes robust pixel-wise correspondence in the cross-modal feature space.

**Human-centric Image Generation**   As illustrated in Figure 4, our model achieves reliable identity preservation even under complex, multi-dimensional control conditions. Guided simultaneously by pose configurations, scene descriptions, and clothing cues, it consistently retains the target character's facial structure and overall appearance across diverse scenes and body configurations. Importantly, ReMix adapts to novel spatial layouts and semantic contexts while avoiding common issues such as identity drift, visual artifacts, and character collapse observed in prior approaches.

**Object-centric Image Generation**   Figure 10 further showcases our model's robustness across diverse scenarios. Even when the subject occupies only a small fraction of the image—such as the digit "3" displayed on an alarm clock interface in the second row—ReMix maintains intricate visual details with high fidelity. This highlights the model's strong capability to preserve semantic and structural consistency, even under spatially constrained or low-resolution conditions.

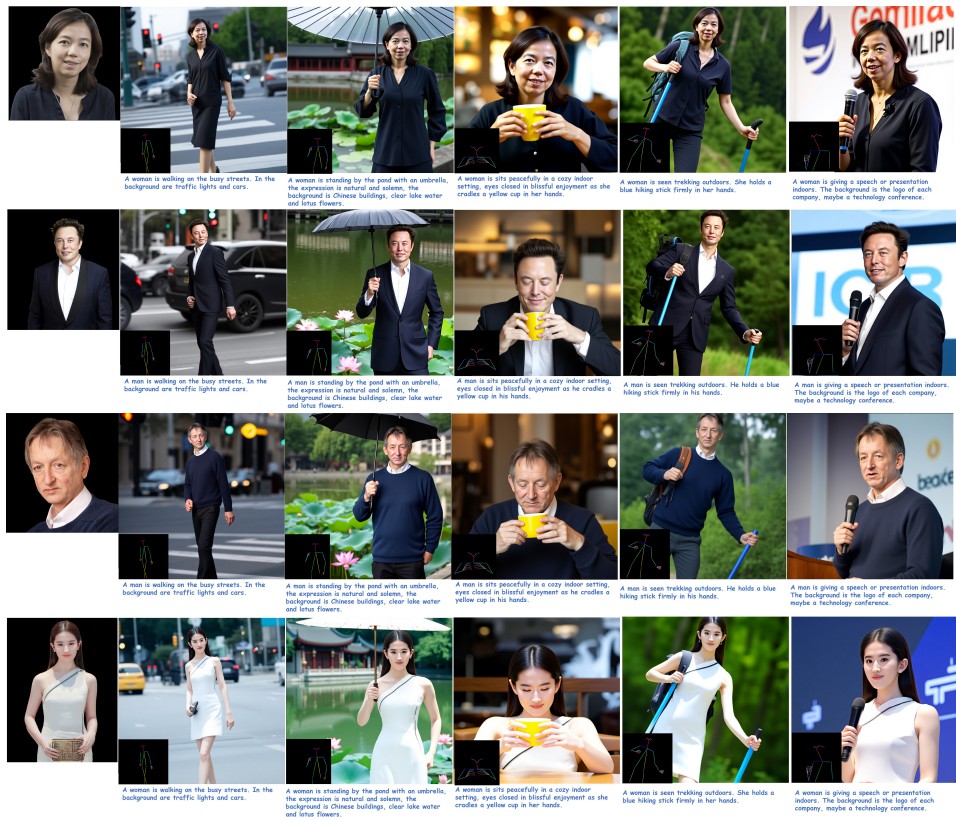

Figure 4: More results for Portrait-Driven Image Generation.

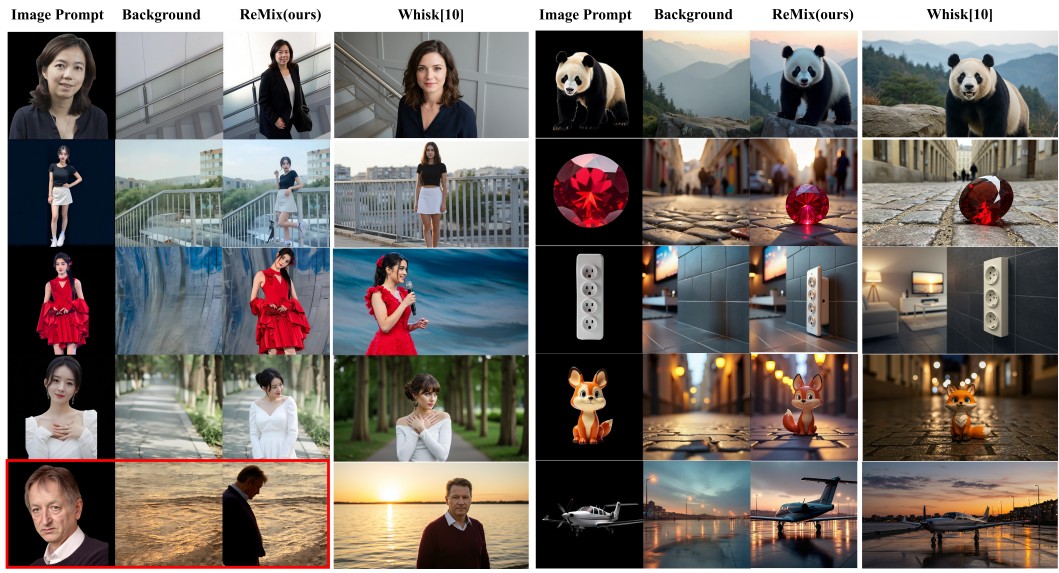

Figure 5: Content-consistent generation results under multiple visual instructions (portrait and background).

## C.2 MULTI-VISUAL-CONDITION GENERATION

Generating images under multiple information-rich visual conditions is particularly challenging due to the inherent tension between strict pixel-level constraints and the stochastic nature of diffusion

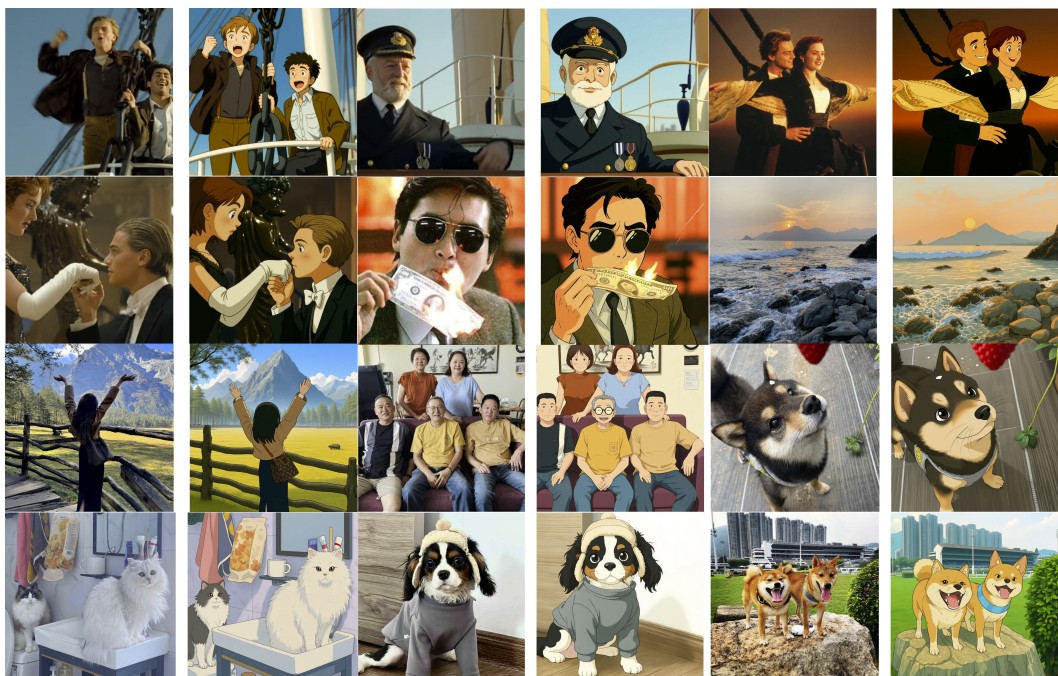

Figure 6: Character consistent Ghibli style transfer.

processes. Existing frameworks often struggle with scalability and flexibility when integrating diverse control signals. Many methods achieve only semantic-level consistency, leaving gaps in fine-grained feature alignment and object-level coherence Google (2024). Our model effectively resolves this conflict. As demonstrated in Figure 5, ReMix outperforms Whisk Google (2024), a closed-source multi-condition generative model, both in multi-condition alignment and perceptual quality. As can be seen, ReMix excels in synthesizing images that satisfy two key requirements: (1) *Context-Aware Blending*, *i.e.*, seamlessly combines disparate visual conditions (*e.g.*, subject + background) while maintaining object consistency; and (2) *Paradox Resolution*, *i.e.*, simultaneously satisfies conflicting requirements through our $\epsilon$-equivariant optimization (highlighted in the red box). Figure 11 show more results in this task.

### C.3 CHARACTER CONSISTENT STYLE TRANSFER

Beyond real-world imagery, our method also generalizes to style transfer tasks with strong identity preservation. We fine-tuned the model using a small set of 200 Ghibli-style image pairs generated by GPT-4o for approximately 2.5k steps. As illustrated in Figure 6, the model successfully captures the artistic style while preserving the original character's visual identity. Furthermore, it demonstrates semantic enhancement abilities, such as rendering richer facial expressions and scene dynamics.

Additionally, we conducted fine-tuning on a limited 3D cartoon dataset. As shown in Figure 7, our method maintains clothing consistency and accurately reproduces fine details like logos on garments, showcasing its fine-grained semantic perception and generalization capabilities across domains.

### C.4 STORY GENERATION

Figure 8 showcases our framework's capability in generating coherent visual narratives across a sequence of images. Given a series of prompts, the model maintains subject identity and scene consistency throughout the progression of the story. This is particularly challenging, as it requires the model to not only preserve character appearance but also adapt to evolving contextual cues such as posture, environment, and emotional expression. Our method effectively bridges these transitions, producing temporally and semantically consistent results, which demonstrates its strong potential

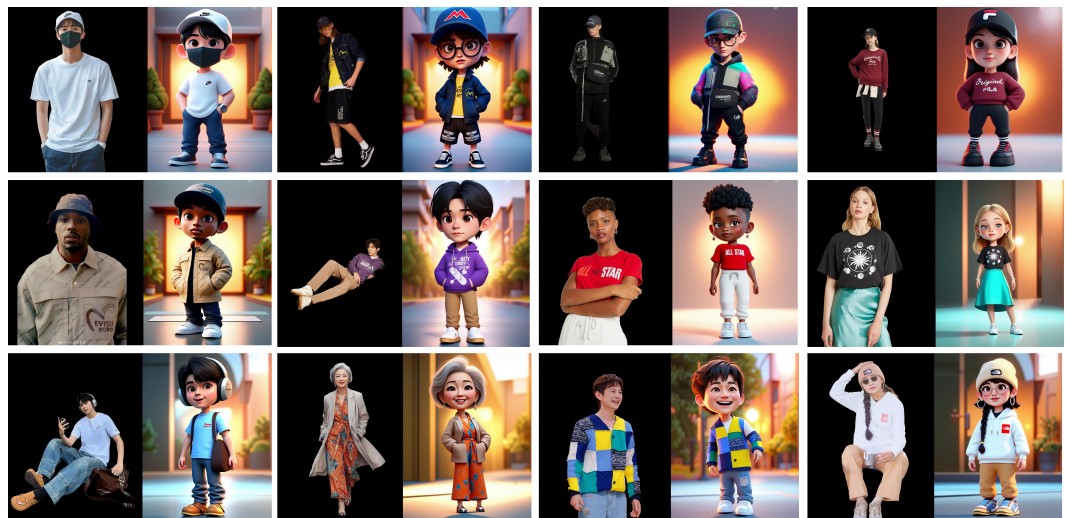

Figure 7: Character consistent 3D cartoon style transfer.

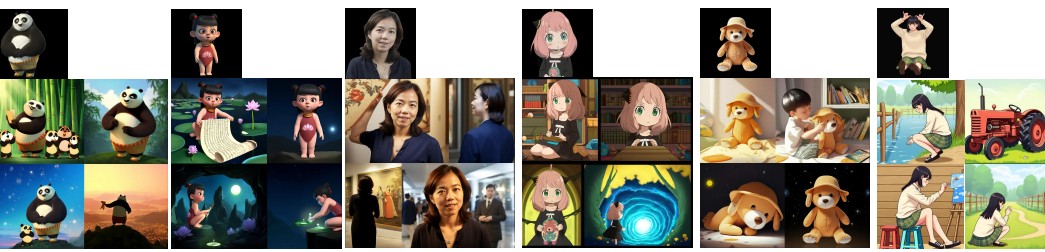

**Story1**: Po teaches a group of eager young animals; Po stands in a tranquil bamboo forest, eyes closed, breathing in fresh mountain air; Po stands atop the tallest peak, facing the rising sun with determination and a playful smile, ready for life's adventures; Po sits on a hill overlooking a valley where village lanterns twinkle like stars.

**Story2**: A girl in a red swimsuit stands on a lotus leaf, holding an open scroll and reading with focused expression; The girl sits inside a cave surrounded by green lotus flowers and water lilies under a bright moon; The girl gently picks a lotus flower with careful, attentive movements; Crouching, the girl touches a lotus flower against a serene night backdrop with stars and distant mountains;

**Story3**: An Asian woman in deep blue stands proudly in a museum, holding a gold-decorated religious painting; Side view shows her elegant profile with tied-up hair and jewelry, expressing calm appreciation; She observes another painting featuring traditionally dressed figures, immersed in cultural history; Front view captures her confident smile, reflecting joy and fulfillment from the artistic experience.

**Story4**: A young girl with pink hair and green eyes is depicted in four scenes from left to right: She is sitting on the floor, engaged in drawing or writing on a piece of paper; She is looking intently at a table, possibly contemplating something; She is holding a teddy bear, standing in a doorway with a curious expression; She is standing alone in a forest, facing a bright, swirling portal of light.

**Story5**: A teddy bear with a hat is shown in various scenes: sitting on a bed, being played with by a child, lying on the ground in a dark setting, and sitting in the dark.

**Story6**: The story of a young girl who finds a small paper boat near a fence, leading her to place her hands in the water and make a wish. She continues her journey through a park, observing a tractor, and finally sitting and painting a picture of a blue sky with clouds and a paper boat. Each scene shows her engagement with different activities, reflecting a peaceful and imaginative day.

Figure 8: Results of Story Generation.

for story-driven content generation and applications in comics, animation, and creative media production.

## C.5 COMPOSITIONAL CONDITIONAL IMAGE GENERATION

An intriguing observation emerged from our compositional conditioning experiments. When constructing reference images by compositing multiple regions—for example, pasting a T-shirt from another source onto the subject—the model still generates seamless outputs that integrate these visual elements naturally, as shown in Figure 9. This implies that our model can robustly interpret and utilize roughly edited or stitched reference inputs without producing visual artifacts. Such flexibility opens up new possibilities for intuitive, user-driven image manipulation. We are actively exploring this capability further to better understand the model's compositional reasoning and push the boundaries of its creative potential.

## D LLM ASSISTANCE STATEMENT

In this paper, we employed ChatGPT-3.5 solely to refine portions of the text for improved readability. Importantly, the model was used only for stylistic adjustments and not for generating original ideas.

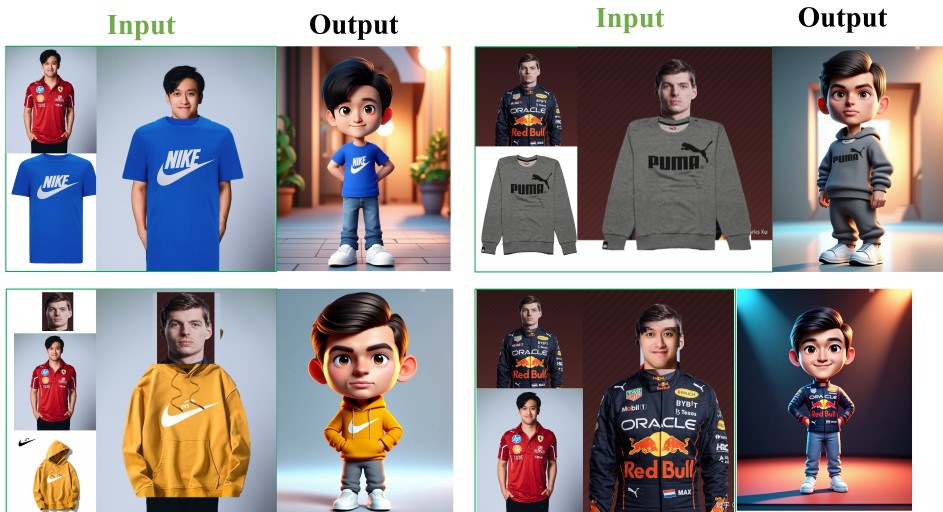

Figure 9: Results of Compositional Conditional Image Generation.

In addition, we used it to help verify the correctness of certain mathematical formulas to minimize the risk of overlooked errors. The research motivation and solutions presented are entirely our own; the large language model did not contribute to the conception or development of these ideas.

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

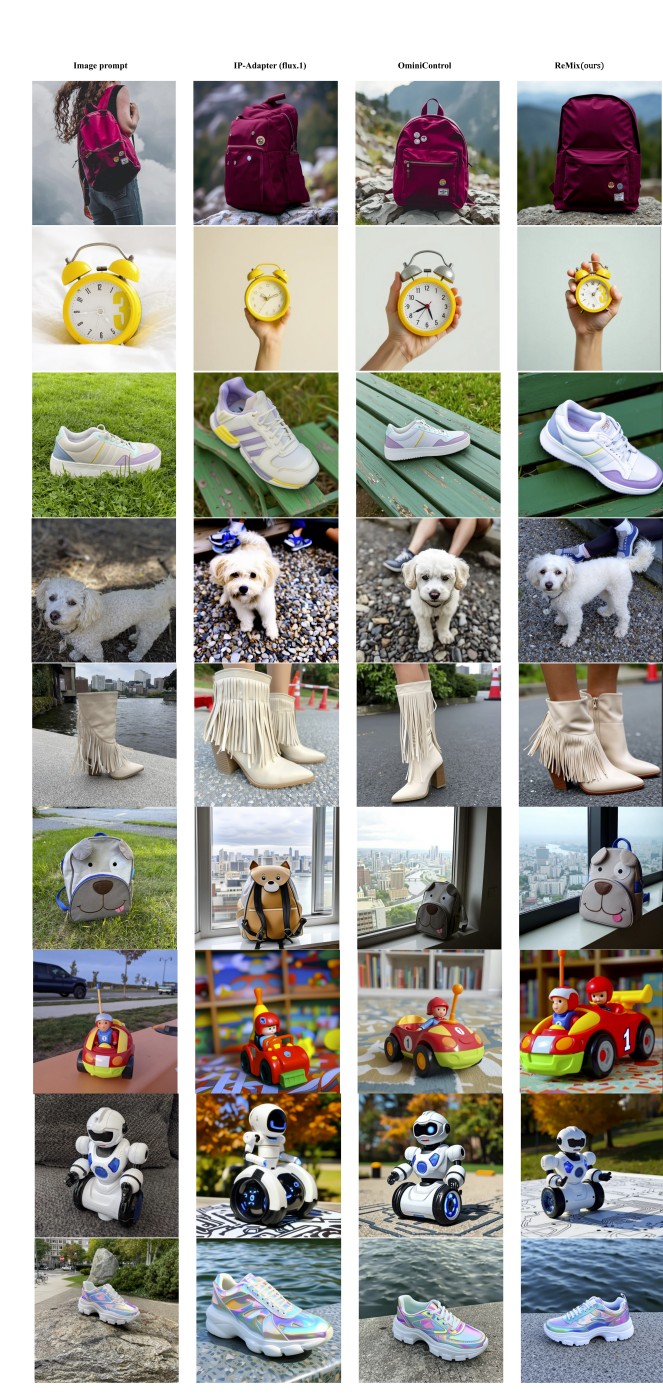

Figure 10: More results for Subject-Driven Image Generation.

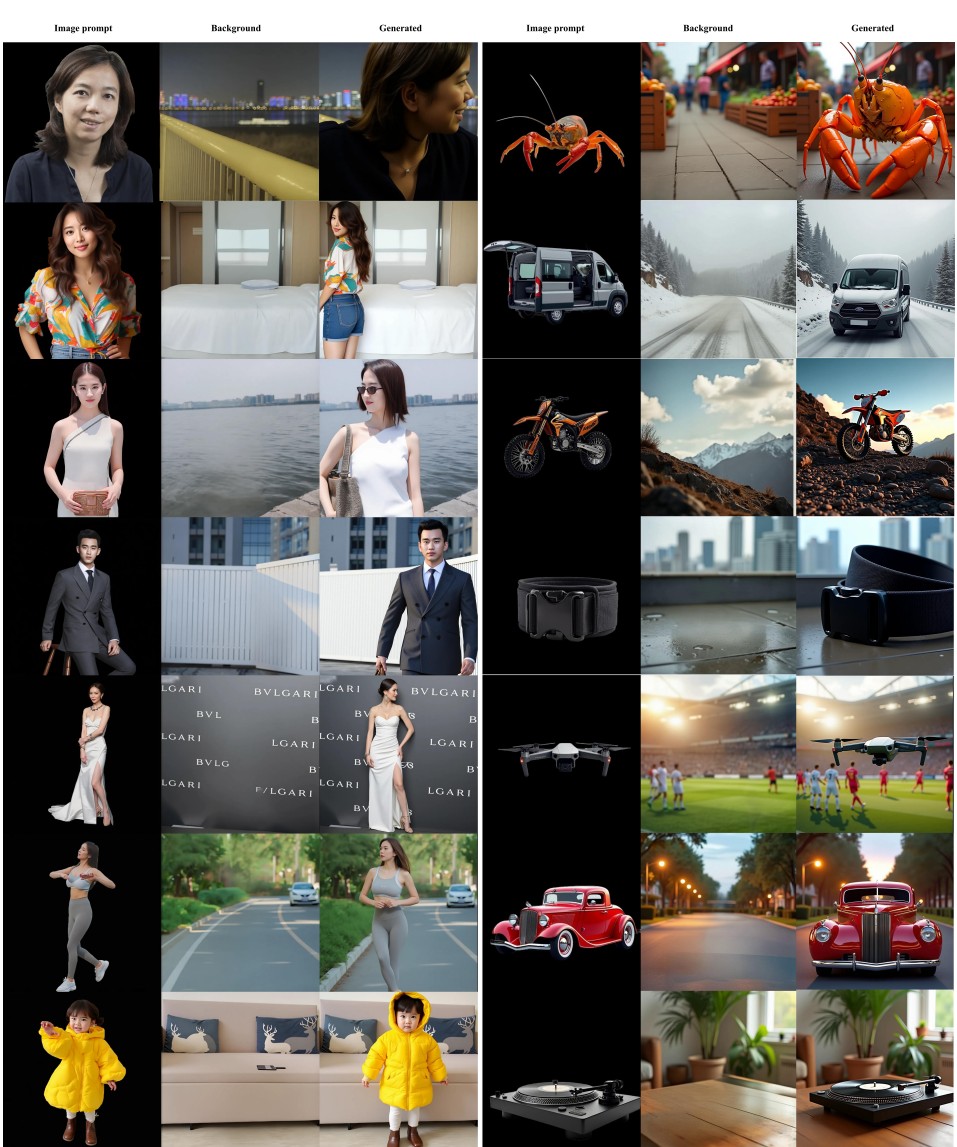

Figure 11: Content-consistent generation results under multiple visual instructions.