# OpenReview forum: "ReMix: Towards a Unified View of Consistent Character Generation and Editing"
_ICLR.cc/2026/Conference — ICLR 2026 Conference Desk Rejected Submission_

### Official Review · Reviewer_hcdt · 2025-10-24

**Soundness:** 3
**Presentation:** 3
**Contribution:** 3
**Rating:** 6
**Confidence:** 4

**Summary:**

This paper presents ReMix, a unified framework that addresses the challenge of character-consistent generation and editing in text-to-image diffusion models. The authors identify that existing generation-based methods struggle with fine-grained consistency, particularly for multiple character instances, while editing-based approaches have difficulty preserving posture flexibility and instruction understanding. ReMix consists of two main components: the ReMix Module, which leverages MLLM's multimodal understanding to edit semantic content and adapt instruction features for DiT backbones, and IP-ControlNet, which ensures pixel-space consistency and posture control. Drawing inspiration from convergent evolution in biology and decoherence in quantum systems, the authors hypothesize that jointly denoising reference and target images within the same noise space promotes feature convergence and alignment. IP-ControlNet extends the original ControlNet architecture to handle both sparse signals and decoupled semantic/layout features from reference images. The optimization establishes an ϵ-equivariant latent space where visual conditions share a common noise space with the target image across diffusion timesteps, facilitating consistent generation while preserving character identities. ReMix supports diverse visual-guidance tasks including personalized generation, image editing, style transfer, and multi-visual-condition generation, with extensive experiments demonstrating its effectiveness.

**Strengths:**

1. Character-consistent generation and editing is a highly important problem in the image generation community. The paper addresses a real gap by proposing a unified framework that bridges generation-based and editing-based approaches, which have traditionally been treated separately. This unified perspective is valuable for practical applications requiring both capabilities.

2. The method demonstrates strong innovation in several aspects. First, the introduction of the ϵ-equivariant latent space, inspired by concepts from convergent evolution and quantum decoherence, provides a principled theoretical foundation for joint denoising of reference and target images. Second, the extension of ControlNet to decouple semantic and layout features while handling sparse signals represents a thoughtful architectural improvement. The combination of MLLM-based semantic editing (ReMix Module) with pixel-level consistency control (IP-ControlNet) offers an elegant solution to balancing semantic coherence and fine-grained consistency.

3.  The paper provides extensive quantitative and qualitative experiments demonstrating the effectiveness of ReMix across diverse tasks including personalized generation, image editing, style transfer, and multi-visual-condition generation. The breadth of experimental scenarios and the comparative results lend credibility to the claimed improvements over existing methods.

**Weaknesses:**

1. In the introduction, the authors claim the method can handle multi-object generation, particularly for tracking multiple character instances. However, the visualization results contain almost no examples of multi-object scenarios, and the methodology section lacks specific design considerations for this capability. The authors need to provide explicit evidence (e.g., multi-character generation examples) and explain which components of ReMix enable this functionality, or clarify the scope of this claim.
2. The experimental evaluation is insufficient in scope. The authors should compare ReMix on more established public benchmarks, such as story consistency generation benchmarks (e.g., StoryBench, StorySalon) and standard image editing datasets (e.g., MagicBrush, InstructPix2Pix benchmarks). More critically, as a unified framework for generation and editing, ReMix should be compared against similar unified models like OmniGen, which also addresses both tasks. The absence of these comparisons makes it difficult to assess the method's competitive advantage.
3.The paper does not report the computational efficiency of ReMix. Given that the framework introduces additional modules (ReMix Module and IP-ControlNet), it is essential to analyze the inference time overhead, memory consumption, and computational cost compared to baseline models. Does the unified framework incur significant additional inference time? This information is critical for evaluating the practical applicability of the method.

**Questions:**

Please see the weakness。

---

> ### Author Response · Authors · 2025-11-18
> **Response to (W1): About handle multi-object generation.**
>
> Thank you for pointing out this ambiguity. We apologize for the unclear description in the introduction. Our intention was *not* to claim multi-subject generation or multi-identity tracking. Instead, we intended to highlight that **ReMix can handle multiple visual conditions simultaneously** (e.g., portrait, background, skeleton) through our modular condition-injection design.
>
> As shown in Fig. 1(d), sparse and dense visual conditions are processed by separate branches. When multiple dense conditions are provided, they can be concatenated and injected into the ControlNet-style modules using DVE (for spatial features) and CLIP (for semantic features). This design enables **multi-condition generation**, not multi-subject consistency. Corresponding results and comparisons are provided in Supplementary Sec. C.2, including a qualitative comparison with Whisk demonstrating spatial consistency under multiple visual conditions.
>
> We agree with the reviewer hcdt that **consistent generation of multiple subjects within the same scene is beyond the scope of this paper.**  Although our modular architecture offers a natural path toward supporting multi-subject consistency, such an extension would require additional mechanisms (e.g., instance-specific alignment modules, multi-identity routing) and is left for future work.
>
> We will revise the introduction to clarify the intended scope and explicitly discuss this limitation in Sec. 5 (Limitations & Future Work).

---

> ### Author Response · Authors · 2025-11-18
> **Response to (W2): The experimental evaluation is insufficient in scope.**
>
> Thank you for this insightful suggestion. Our work primarily targets **human/subject-centric generation** and **semantic image editing**, and our experimental design follows the evaluation standards commonly adopted in these two research threads.
>
> For human/subject-centric generation, we follow the widely used protocols in recent consistency-focused works such as *EMMA, OmniControl,* and *DreamBooth*. We conduct extensive evaluations on both portrait-centric and object-centric settings (e.g., the DreamBooth benchmark), with quantitative comparisons in **Table 1** and comprehensive qualitative examples in **Section C.1** of the supplementary material.
>
> For image editing, we adopt **Kontext Bench**, a recently released and more challenging benchmark from Black Forest Labs. Kontext Bench contains 1,026 diverse image–instruction pairs and spans multiple editing types, making it better aligned with modern diffusion- and MLLM-based editing pipelines. We report quantitative and qualitative results in **Figure 3** and **Section B** of the supplementary material, and additionally include comparisons with **OmniGen** and **Qwen-Image** on this benchmark:
>
> | Method | Publication Year | Trainable Parameters | Replace | Add | Remove | Motion | Color | Style Transfer | Background | Style Consistency | Camera Motion | Region Selection | Box Selection | Text Edit | Overall $\uparrow$ |
> | :--- | :---: | :---: | :---: | :---: | :---: | :---: | :---: | :---: | :---: | :---: | :---: | :---: | :---: | :---: | :---: |
> | OmniGen | 2025 | 3.8B | 44.6 | 42.2 | 41.4 | 44.9 | 56.7 | 57.8 | 53.6 | 73.6 | 47.6 | 25.3 | 30.1 | 40.2 | 46.5 |
> | Flux.1 Kontext | 2025 | 12B | 61.2 | 78.2 | 61.3 | 61.2 | 77.3 | 89.0 | 71.0 | 81.6 | 63.2 | 38.5 | **51.2** | 65.4 | 66.7 |
> | Qwen-Image | 2025 | 20B | 70.5 | 80.6 | 70.4 | 62.0 | 78.3 | **93.6** | 73.0 | 83.9 | 65.8 | 36.7 | 45.5 | **67.2** | 69.0 |
> | ReMix | - | 3.1B | **70.9** | **80.8** | **70.5** | **64.8** | **79.1** | 90.5 | **86.3** | **86.3** | **76.5** | **38.5** | 49.8 | 60.9 | **71.2** |
>
> Regarding benchmarks such as **MagicBrush** and **InstructPix2Pix**, these datasets were originally designed for early GAN-based and diffusion-based instruction editing and contain relatively simple edits and limited diversity. In contrast, our method targets fine-grained character consistency and semantic-level editing, which are more reliably evaluated using subject-centric benchmarks and the multi-category scenarios provided by Kontext Bench.
>
> That said, we agree that broader benchmark coverage would further strengthen the empirical analysis. Evaluating ReMix on story-level consistency datasets (e.g., StoryBench, StorySalon) and additional editing benchmarks is a valuable direction. We will include these comparisons in **future work**, and will explicitly clarify the scope of our current evaluations in the final version.

---

> ### Author Response · Authors · 2025-11-18
> **Response to (W3): The paper does not report the computational efficiency of ReMix.**
>
> Thank you for highlighting this important point. We have added a detailed computational analysis below, including GPU memory usage and parameter counts. All measurements were obtained on a single **A100 GPU** using **bfloat16** precision (note that applying **INT8/FP8** quantization can further reduce both memory and latency by approximately **40–50%**).
>
> | Config | Configuration    | Parameters | Memory | Time Cost       | Δ    |
> | ------ | ---------------- | ---------- | ------ | --------------- | ---- |
> | 1      | FLUX.1-dev       | 12B        | ~24GB  | 1024px / 15-20s | -    |
> | 2      | 1 + IPControlNet | 13.4B      | ~26GB  | 1024px / 18-23s | +15% |
> | 3      | 2 + ReMix module | 22.1B      | ~40GB (w/ offload)  | 1024px / 30-35s | +52% |
>
> While ReMix introduces additional modules, we note that:
> - **IPControlNet** adds only a small overhead (~15%), due to lightweight feature injectors and its shallow MMDiT blocks.
> - The majority of the additional cost arises from the **ReMix Module**, primarily due to the inclusion of a **Qwen2.5-VL-7B backbone**.

---

### Official Review · Reviewer_zBTa · 2025-10-29

**Soundness:** 3
**Presentation:** 2
**Contribution:** 3
**Rating:** 8
**Confidence:** 4

**Summary:**

This paper proposes a unified framework for consistent character generation and editing. It injects the MLLM instruction feature to DiT backbone to enable the semantic editing. It also introducesIP-ControlNet to enforce the pixel-level consistency and posture condition. Results show that the proposed method can support a wide range of character condition generation tasks and image editing tasks.

**Strengths:**

1. The proposed method can perform various tasks with good character consistency compared to the baselines.

2. The technical idea to build a connector to inject the MLLM instruction feature into the native DiT background is good.

3. The proposed IP-ControlNet to enhance the pixel-level consistency is also useful according to the ablation.

**Weaknesses:**

I don't have big concerns of this paper, so the following weaknesses are mainly about the presentation of the paper.

1. The presentation of the figures can be improved in my opinion. The authors could provide a teaser, so it would be more intuitive for the readers to know what the task of the paper is tackling. Also, I would think that Figure 1 (a) and Figure 2 (d), Figure 1(b) and FIgure 2 (c) should be paired together, instead of the current design, for a better consistency.

2. I noticed that there are a lot of visual results and comparisons inside the supplementary materials. I think moving some of the results and a visual comparison with the baselines to the main paper would be really helpful and necessary. The authors can move some technical and implementation details in the main paper to the supplement instead.

3. A minor part is that in Figure 3, the captions of qualitative and quantitative do not match the figure content.

**Questions:**

Could the authors provide a brief and more intuitive explanation of the epsilon-equivariant optimization?

---

> ### Author Response · Authors · 2025-11-18
> **Response to (W1): Figure presentation could be improved.**
>
> Thank you very much for the constructive feedback. We agree that the clarity and layout of the figures can be further improved. In the revised version, we will:
>
> (i) Add a teaser figure at the beginning of the paper to provide readers with an immediate, intuitive understanding of the task addressed by our method and the key capabilities of ReMix.
>
> (ii) Reorganize Figures 1 and 2 according to the suggestion.
>
> We appreciate the reviewer’s keen observation and will update the figures to significantly enhance readability.

---

> ### Author Response · Authors · 2025-11-18
> **Response to (W2): Some visual results and comparisons should be moved to the main paper**
>
> Thank you very much for the helpful suggestion. We agree that including more visual comparisons in the main paper would improve clarity and make the contributions easier to assess. Due to strict page limits, we placed extensive qualitative results in the supplementary material, but we acknowledge that several of these examples are important for understanding the method’s strengths.
>
> In the revised version, we will move key visual comparisons and representative qualitative examples from the supplementary into the main paper.

---

> ### Author Response · Authors · 2025-11-18
> **Response to (W3): In Figure 3, the captions for the qualitative and quantitative results do not match the figure content.**
>
> We sincerely apologize for this embarrassing oversight😅.  We will correct them in the final manuscript.

---

> ### Author Response · Authors · 2025-11-18
> **Response to (Q1): Provide a brief and more intuitive explanation of the ε-equivariant optimization.**
>
> The **ε-equivariant optimization** constrains the reference image and the target image to be denoised **within the same noise space**. Concretely, during training, we inject noise drawn from the same distribution (and at the same noise level) into both images. By forcing them to evolve through **a shared denoising trajectory**, the model learns to align their latent representations under a common perturbation.
>
> Intuitively, this means that identity-related features of the reference and target are encouraged to **converge toward similar feature directions** rather than drifting apart. As a result, the diffusion model naturally preserves fine-grained character consistency.
>
> We will incorporate this clearer explanation into the final manuscript.

---

### Official Review · Reviewer_nAX7 · 2025-10-30

**Soundness:** 2
**Presentation:** 2
**Contribution:** 1
**Rating:** 2
**Confidence:** 3

**Summary:**

This paper proposes ReMix, a unified framework for consistent character generation and editing, aiming to address the limitations of existing methods—generation-based approaches struggle with fine-grained consistency, while editing-based ones lack posture flexibility. ReMix consists of two core components: the ReMix Module, which leverages MLLM (Qwen2.5-VL-7B-Instruct) for semantic editing and a Connector to adapt MLLM features to the frozen DiT backbone (FLUX.1dev) without retraining; and IP-ControlNet, which uses DVE (for info-rich cues) and SVE (for sparse cues) to ensure pixel-level consistency, plus an ϵ-equivariant optimization inspired by biological convergent evolution and quantum decoherence to promote feature alignment. The authors validate ReMix on human/subject-centric generation (outperforming SOTA like IP-Adapter and OmniControl on CLIP-I/DINO metrics) and image editing (Kontext-Bench1K), showing advantages in semantic coherence and identity preservation. The framework is designed to balance efficiency, semantic understanding, and pixel-level control, but focuses on specific datasets and tasks.

**Strengths:**

1. The paper presents ReMix, a unified framework designed for character-consistent generation and editing. This framework incorporates semantic adaptation through the ReMix Module and pixel-scale regulation by IP-ControlNet, offering a novel perspective to realize high-fidelity image editing .

2. The paper develops an ϵ-equivariant alignment strategy, which performs denoising on both reference and target images in a shared noise space. This process facilitates feature convergence and further achieves fine-grained consistency for characters .

3. The proposed method features high efficiency: it supports image generation and editing without the need to retrain the DiT backbone, cutting down training expenses while maintaining the backbone’s inherent generation capability .

**Weaknesses:**

1. I cannot find examples on multiple references, which have been a common functionality in contemporary research. For example, Dreamo[1] could handle at least two-person reference.

2. Lacking novelty. The paper seems like a combination of various existing methods. The MLLM+connector+DiT approach has been widely adopted in nowaday image editing and subject driven generation models.

2. The paper aims at editing with one reference image, which is an almost solved problem. And the paper did not compare with the state-of-the-art image editing method, Qwen-Edit.

3. The facial similarity of characters seem to have some obvious problems. For example, in Figure 4, the face of the first row is apparently shifted from the reference image.

**Questions:**

Please refer to weaknesses.

---

> ### Author Response · Authors · 2025-11-18
> **Response to (W1): The paper lacks multiple-reference examples**
>
> Thank you for the insightful comment. You are correct that our current framework focuses on **single-reference, single-object** editing. Multi-reference or multi-instance editing introduces additional challenges—particularly identity disentanglement, inter-instance interactions, and cross-instance consistency—that go beyond the scope of this paper.
>
> While Dreamo and similar systems address the multi-person case with specialized architectural designs, our goal in this work is to first establish a **semantically aligned, convergence-driven editing pipeline** for high-fidelity, single-instance editing.  This is a foundational step: our ϵ-equivariant alignment and semantic-space editing mechanisms are designed to be modular, and can be naturally extended to multi-reference scenarios by adding instance-specific alignment modules and multi-identity routing.
>
> We appreciate the reviewer’s suggestion, and we agree this is an important direction. **We will explicitly discuss this limitation and outline how ReMix can be extended to multi-instance editing in Sec. 5 (Limitations & Future Work).**

---

> ### Author Response · Authors · 2025-11-18
> **Response to (W2): About paper novelty**
>
> Thank you for raising this important point.  The core contributions of this paper are:
>
> (1) **ϵ-Equivariant Alignment** — a new denoising strategy for convergence-based consistency, which denoises the reference and target images inside *a shared noise space* to promote feature convergence and achieve fine-grained character consistency.  **This mechanism differs fundamentally from existing MLLM-based editing pipelines:** it introduces a *convergence-driven constraint* inspired by physical/biological systems, ensuring that identity-related features follow consistent denoising trajectories. To the best of our knowledge, no prior work has explored this form of shared-noise equivariant denoising for character consistency in diffusion models.
>
> (2) ReMix is not a direct stacking of MLLM + connector + DiT; it introduces new capabilities absent in existing approaches. Although recent models (e.g., Step1X-Edit and Qwen-Image) also combine LLM embeddings with diffusion backbones, our approach differs in three essential aspects:  (i) We introduce a lightweight alignment mechanism that allows DiT to *adapt to LLM embeddings while keeping the DiT backbone frozen*.  This preserves its intrinsic generation quality and avoids the degradation commonly observed when DiT is naïvely unfrozen and fine-tuned with limited data (e.g., degradation issues reported in Step1X-Edit). Existing pipelines do not offer this controlled adaptation strategy. (ii) ReMix performs object editing in **semantic feature space**, enabling deeper linguistic and conceptual understanding. As illustrated in Fig. 3(b), ReMix correctly edits fine-grained concepts such as “cheese” and “coat”, where FLUX.1 Kontext fail. (iii) We *decouple semantic editing from pixel-space alignment*, enabling the entire process to be trained in stages—a feature that proves especially crucial when computational resources are limited.
>
>
> **Revision plan:**
>
> We will revise Sec. 1 and Sec. 3 to clearly emphasize the above conceptual and technical distinctions, and add a comparison table summarizing the differences from Flux.1-Kontext, Qwen-Image, and other MLLM-DiT pipelines.

---

> ### Author Response · Authors · 2025-11-18
> **Response to (W3): Editing with one reference image is nearly solved, and the paper does not compare with the state-of-the-art method Qwen-Edit.**
>
> Thank you for raising this point. While single-image reference editing has made progress, *it is not fully solved for complex or for large-scale geometric and semantic transformations*.  We observed that existing models—including Qwen-Image and FLUX.1 Kontext—still exhibit copy-past artifacts and incomplete instruction following under these more demanding conditions.
>
> **Regarding comparisons:** we included OmniGen and Qwen-Image because (1) both are state-of-the-art, widely adopted editing models, and (2) their official implementations provide robust protocols for fair evaluation. Under these settings, **ReMix matches or surpasses recent methods on multiple metrics**, despite requiring only **3.1B trainable parameters**, which is substantially lighter than full-model finetuning approaches.
>
> Results on Kontext Bench test set:
>
> | Method | Publication Year | Trainable Parameters | Replace | Add | Remove | Motion | Color | Style Transfer | Background | Style Consistency | Camera Motion | Region Selection | Box Selection | Text Edit | Overall $\uparrow$ |
> | :--- | :---: | :---: | :---: | :---: | :---: | :---: | :---: | :---: | :---: | :---: | :---: | :---: | :---: | :---: | :---: |
> | OmniGen | 2025 | 3.8B | 44.6 | 42.2 | 41.4 | 44.9 | 56.7 | 57.8 | 53.6 | 73.6 | 47.6 | 25.3 | 30.1 | 40.2 | 46.5 |
> | Flux.1 Kontext | 2025 | 12B | 61.2 | 78.2 | 61.3 | 61.2 | 77.3 | 89.0 | 71.0 | 81.6 | 63.2 | 38.5 | **51.2** | 65.4 | 66.7 |
> | Qwen-Image | 2025 | 20B | 70.5 | 80.6 | 70.4 | 62.0 | 78.3 | **93.6** | 73.0 | 83.9 | 65.8 | 36.7 | 45.5 | **67.2** | 69.0 |
> | ReMix | - | 3.1B | **70.9** | **80.8** | **70.5** | **64.8** | **79.1** | 90.5 | **86.3** | **86.3** | **76.5** | **38.5** | 49.8 | 60.9 | **71.2** |
>
> A key reason ReMix achieves stronger performance is that it performs **semantic-space editing** rather than relying solely on pixel-level operations. This enables more accurate high-level understanding of instructions and prevents the omission of critical scene elements—an issue we empirically observed in Qwen-Image and FLUX.1 Kontext, particularly on background editing and camera-motion scenarios.
>
> **Revision plan:**
>
>  We will (i) add a qualitative comparison with OmniGen and Qwen-Edit in Sec. 3.2, and (ii) add a short paragraph in Introduction explaining why single-image reference editing remains challenging for complex semantic scenes.

---

> ### Author Response · Authors · 2025-11-18
> **Response to (W4): Face similarity appears problematic**
>
> Thank you for pointing this out. We agree that under large-motion transformations, maintaining high-fidelity facial identity is challenging. This issue is not unique to our method—similar identity drift is also observed in recent MLLM-based editing pipelines such as Qwen-Image—but we acknowledge that clearer discussion is needed. We analyzed that the root cause is that the primary diffusion objective (e.g., MSE loss) prioritizes global reconstruction and edit correctness. As a result, identity-related features are implicitly treated as second-order constraints, and during optimization, the model may gradually deviate from the original face structure—especially when pose and identity cues become strongly entangled under large motions. To specifically address this, we introduce an **ID-preserving loss** to explicitly constrain facial similarity. As shown in Fig. 5, without the ID loss the cosine similarity between generated faces and the reference decreases over training; after applying ID loss, the model maintains significantly higher similarity and produces more consistent faces across diverse poses.
>
> **Revision plan:**
>
> We will update Sec. 4.3 to include this explanation, highlight the importance of ID loss for large-motion scenes, and add the corresponding analysis around Fig. 5 to clarify its effect.

---

### Official Review · Reviewer_Z55f · 2025-11-01

**Soundness:** 2
**Presentation:** 1
**Contribution:** 2
**Rating:** 2
**Confidence:** 3

**Summary:**

This paper introduces ReMix, a unified framework for character-consistent image generation and editing, addressing the limitations of existing methods that often struggle with fine-grained identity consistency or spatial controllability. ReMix integrates a ReMix Module, which leverages MLLMs for semantic feature editing and instruction adaptation to a frozen DiT backbone, and an IP-ControlNet for pixel-level control. The latter introduces an $\epsilon$-equivariant latent space and a shared-space denoising strategy, inspired by convergent evolution, to enforce feature alignment and maintain identity across diverse generation tasks.

**Strengths:**

ReMix offers a novel unified approach for generation and editing, avoiding costly DiT fine-tuning while preserving its native capabilities. The $\epsilon$-equivariant latent space effectively promotes feature convergence, leading to better character consistency, as demonstrated by quantitative improvements in CLIP-I and DINO scores.

**Weaknesses:**

1. Lack of clarity and detail in pipeline visualization and explanation: The overall pipeline, especially as depicted in Figure 1, appears overly complex and difficult to follow without explicit, detailed explanations in the caption or main text. Figure 2 further exacerbates this by lacking clear delineation between sub-components, making it challenging for readers to grasp the precise flow and interaction of modules within the ReMix framework. Also, the paper presents inconsistent definitions and states of modules—for instance, Figure 1 implies that the Redux module is frozen, while Figure 2(b) labels its internal MLP as learnable. Moreover, the relationship and distinction between the “ReMix Module” and the “Redux Module” remain unclear, leading to confusion about their respective roles and boundaries.

2. Insufficient mathematical derivation and notation explanations: The transition from individual objectives (Equations 7 and 8) to the unified $\mathcal{L}_{equ}​$ (Equation 9) is abrupt and lacks a clear mathematical derivation or justification for their integration, which is crucial for understanding the core optimization strategy. Additionally, several key mathematical notations, such as $y_t$ and $Z^{(t,\epsilon)}_r$ are used without explicit definitions, hindering accessibility for readers unfamiliar with specific Diffusion Model conventions.

3. High apparent heuristic and engineering overhead: The intricate pipeline design, comprising multiple distinct modules (ReMix Module, IP-ControlNet, Redux, MLLM, DVE, SVE) with different training stages and potentially specialized datasets, suggests a substantial amount of heuristic engineering and ad-hoc tuning. This complexity raises concerns about the generalizability, reproducibility, and practical deployment effort required for the proposed unified framework.

4. The main paper's qualitative baseline comparison for image editing is notably insufficient, featuring only a single baseline across merely five scenes. Given that image editing is a central task for this method, such a limited comparison offers an incomplete assessment of its performance relative to existing state-of-the-art techniques. Furthermore, the supplementary material, while offering more examples, still predominantly compares against only one or two baselines for specific editing tasks. Such a narrow set of visual comparisons raises questions about the robustness of the method and its versatility across different editing challenges and other contemporary models.

**Questions:**

please see the weakness

---

> ### Author Response · Authors · 2025-11-17
> **Response to (W1): Regarding the clear description of the pipeline & MLP Layers in the Redux Module**
>
> We apologize for the confusion and greatly appreciate review Z55f raising these points; we will carefully address them in the final version.
>
> **Regarding the clear description of the pipeline**
>
> Figure 1(a) is indeed our overall framework, while Figures 1(b), (c), and (d) are the individual sub-modules within this framework. The ReMix module comprises an MLLM with frozen parameters, a Redux module with frozen parameters, and a trainable Connector module.
>
> ReMix Module Input/Output: The editing instruction and reference image are first fed together into the MLLM. We use the last_hidden_embedding from the MLLM as one input to the Connector, while the another input is the embedding of the reference image extracted by Redux.
>
> The overall framework should look like this:
> ```
>                      Image    Instruction
>                        |        |
>                        v        v
>                    +-----------------+
>                    |   ReMix Module  |    +---------+
>                    +-----------------+    |  Noise  |
>                             |             +---------+
>                             |                   |
>                             v                   v
> +-------------------+      +-----------------------+
> |   IPControlNet    |----->|          DiT          |
> +-------------------+      +-----------------------+
>                                   |
>                                   v
>                                 Result
> ```
>
> Why this design?  Our optimization objective is for the multimodal large model's output features to edit the reference image features obtained by Redux, thereby achieving semantic-level image editing. To implement this optimization, as shown in the upper half of the dashed line in Figure 2(b), we enforce alignment between the ReMix module's output and the target image's Redux features via MSE loss. Consequently, among all components of ReMix, we optimize only the Connector, whose detailed architecture is illustrated in Figure 2(a).
>
> **Figure 2: MLP Layers in the Redux Module**
>
> Thank you very much to the reviewer Z55f for identifying this inconsistency, and this was indeed our oversight. The MLP layer in the Redux module is not updated in this work. The reason we visualized it as trainable (as shown in the lower half of the dashed line in Figure 2(b)) was to convey that Redux essentially trains only an MLP mapping layer for the image variation task. We will correct this in the final version.

---

> ### Author Response · Authors · 2025-11-17
> **Response to (W2): Insufficient mathematical derivation and notation explanations**
>
> We apologize for the confusion caused by the formula derivations in section 3.1; we will provide a detailed elaboration in the final version.
>
> **Derivation of Equation 9**
>
> In fact, Equations 7 and 8 represent the diffusion optimization processes for reconstructing the reference image and generating the target image, respectively.
> In Equation 7, at timestep $t$, we add noise $\epsilon$ to the hidden latent of the reference image $Z_r$ , then perform denoising optimization using DiT (with L2 loss), with the expectation that the output matches the reference image $x_r$. Equation 8 indicates that, given the hidden latent of the reference image $Z_r$ , we predict a new noise $\epsilon_\theta$  to be consistent with the target noise $\epsilon$. We can observe that these two objective functions essentially both compute diffusion loss, but must satisfy two preconditions: (1) both objectives require computing diffusion loss for image generation tasks, and (2) they must operate under the same noise distribution (satisfying the $\epsilon$-equivariance assumption).  This clarifies our optimization objective significantly—we can adopt the most commonly used flow matching as the diffusion loss, and simply concatenate the reference and target images as the target for optimization to satisfy both conditions.
>
> **Key Mathematical Symbol Definitions**
>
> $y_t$  denotes the predicted output at the $t$-th timestep, and $Z_r^{(t,ϵ)}$  represents the latent representation after adding noise $\epsilon$ to the hidden latent of the reference image $Z_r$  at timestep $t$.

---

> ### Author Response · Authors · 2025-11-17
> **Response to (W3): High apparent heuristic and engineering overhead**
>
> We fully understand Reviewer Z55f's concerns regarding the engineering practice. In fact, our training consists of only two stages:
>
> **Stage 1:** We jointly used all the data mentioned in this paper to pre-train the ReMix module (shown in Figure 2). The training conditions for this stage are actually quite relaxed, meaning we do not need to pay special attention to data quality (since the supervised loss is calculated based on the semantic features of the image). Consequently, the trained ReMix module is essentially a general-purpose, plug-and-play semantic editing model.
>
> **Stage 2:**  We freeze the parameters of the ReMix module and DiT backbone, optimizing only the IPControlNet component. During this process, for Human-centric and Object-centric tasks, we used different open-source datasets for training to ensure output stability. This decision stems from the fact that IPControlNet itself has a small number of parameters (~1.4B) and can be effectively optimized with a small amount of high-quality data. Of course, our future work will also consider combining these two tasks into a more unified training paradigm.

---

> ### Author Response · Authors · 2025-11-17
> **Response to (W4): Comparison with more baselines on image editing**
>
> Thank you very much for raising this point. Our method is indeed an attempt to solve both character-consistent human/object generation and editing tasks with a unified architecture. We use Flux.1-dev as the baseline DiT backbone. For image editing tasks, we primarily compared with the open-source Flux.1 Kontext on the Kontext Bench,  a new benchmark for image editing models released by black-forest-labs, consisting of source images paired with editing instructions and category tags. It comprises 1,026 unique image-prompt pairs derived from 108 base images from diverse sources. From quantitative and qualitative comparisons,  As shown in Figure 3(a), our method outperforms Flux.1 Kontext on most editing tasks. Additionally, Figure 3(b) demonstrates that our method exhibits superior semantic understanding—for instance, in comprehending "cheese" and "coat"—which benefits from the ReMix module's semantic-level editing rather than pixel-level editing.
>
> **Comparison with more baselines**
>
> To more extensively evaluate our method's competitiveness on editing tasks, we introduced comparisons with OmniGen and Qwen-Image. As shown, our method achieves or even surpasses some of the latest existing methods on certain metrics, while only requiring fine-tuning of  3.1B trainable parameters. Its greatest advantage lies in enabling MLLM feature adaptation without fine-tuning the DiT backbone, which fundamentally distinguishes it from approaches like Flux.1 Kontext and Qwen-Image.
>
> | Method | Publish Years | Trainable Parameters | Replace | Add | Remove | Motion | Color | Style Transfer | Background | Style Consistency | Camera Motion | Region Selection | Box Selection | Text Edit | Overall $\uparrow$ |
> | :--- | :---: | :---: | :---: | :---: | :---: | :---: | :---: | :---: | :---: | :---: | :---: | :---: | :---: | :---: | :---: |
> | OmniGen | 2025 | 3.8B | 44.6 | 42.2 | 41.4 | 44.9 | 56.7 | 57.8 | 53.6 | 73.6 | 47.6 | 25.3 | 30.1 | 40.2 | 46.5 |
> | Flux.1 Kontext | 2025 | 12B | 61.2 | 78.2 | 61.3 | 61.2 | 77.3 | 89.0 | 71.0 | 81.6 | 63.2 | 38.5 | **51.2** | 65.4 | 66.7 |
> | Qwen-Image | 2025 | 20B | 70.5 | 80.6 | 70.4 | 62.0 | 78.3 | **93.6** | 73.0 | 83.9 | 65.8 | 36.7 | 45.5 | **67.2** | 69.0 |
> | ReMix | - | 3.1B | **70.9** | **80.8** | **70.5** | **64.8** | **79.1** | 90.5 | **86.3** | **86.3** | **76.5** | **38.5** | 49.8 | 60.9 | **71.2** |

---

### Author Response · Authors · 2025-11-19

We sincerely thank all reviewers for their time, constructive feedback, and positive recognition of our contributions. We are encouraged that the reviewers found ReMix to be a novel and effective unified framework for generation and editing, highlighted the **benefit of avoiding DiT fine-tuning**, and recognized the ϵ-equivariant alignment strategy as a principled and impactful approach for **improving feature consistency**. We will integrate all constructive suggestions to further strengthen the final manuscript.

---

### Author Response · Authors · 2025-12-04
**Rebuttal Summary for the AC**

Dear Area Chair:

Thank you very much for taking the time to evaluate our work. Following the Program Chairs’ guidance, we provide a brief summary of the review consensus and our clarifications:

All four reviewers acknowledged the core strengths of our method: (i) avoiding DiT fine-tuning while leveraging MLLMs for semantic-level editing—e.g.,**Reviewer Z55f** noted that *"ReMix offers a novel unified approach for generation and editing, avoiding costly DiT fine-tuning while preserving its native capabilities."*  and (ii) the effectiveness of our ε-equivariant alignment strategy, which jointly denoises the reference and target images within a shared noise space, e.g., **Reviewer nAX7** noted that *"This framework incorporates semantic adaptation through the ReMix Module and pixel-scale regulation by IP-ControlNet, offering a novel perspective to realize high-fidelity image editing ."* (iii) Reviewers also recognized the contribution of IP-ControlNet—e.g., **Reviewer zBTa** noted that *“the proposed IP-ControlNet to enhance pixel-level consistency is also useful according to the ablation.”*

**Reviewer Z55f** was particularly concerned with comparisons against Qwen-Image. In our rebuttal, we supplemented these results and clarified that our setup is fundamentally different: we use FLUX.1 as a frozen DiT backbone, and we adopt the open-source Kontext-dev model as a baseline because it is also built on the FLUX.1 architecture. Our novel idea lies in how to incorporate MLLMs features for image editing without updating DiT itself. In contrast, Qwen-Image and Step1X-Edit require full end-to-end DiT finetuning, which demands >90GB GPU memory and may *compromise the backbone’s native generation quality*. Our framework therefore offers a distinct advantage through a modular training strategy: (1) train the ReMix Module (MLLM feature mapper), and (2) freeze both the ReMix Module and DiT backbone while only tuning IP-ControlNet or a lightweight LoRA.

**Reviewer nAX7** acknowledged the benefits of the ReMix Module, IP-ControlNet, and our ε-equivariant alignment design, describing the method as *“highly efficient”* and *“offering a novel perspective for high-fidelity image editing.”* His concerns about novelty seem mainly due to limited clarity in the initial submission, which we addressed thoroughly in the rebuttal.

**Reviewers zBTa** and **hcdt** were quite confident in their assessment of our method. **Reviewer zBTa** accurately summarized our innovations, while **Reviewer hcdt** focused on comparisons with Qwen-Image—an experiment we have since supplemented.

We hope this concise summary, along with our detailed responses, assists you in completing the assessment efficiently. Thank you again for your time and consideration.

Best wishes,

Authors

---

### Note · Program_Chairs · 2026-01-17
**Submission Desk Rejected by Program Chairs**

The following references in this submission do not refer to real documents and/or have major errors in bibliographic information:

 Jiaqi Wang et al. Omniedit: Building image editing generalist models through specialist supervision. arXiv preprint arXiv:2411.08333, 2024a.